# Extend Model Merging from Fine-Tuned to Pre-Trained Large Language Models via Weight Disentanglement

## Abstract

Merging Large Language Models (LLMs) aims to amalgamate multiple homologous LLMs into one with all the capabilities. Ideally, any LLMs sharing the same backbone should be mergeable, irrespective of whether they are Fine-Tuned (FT) with minor parameter changes or Pre-Trained (PT) with substantial parameter shifts. However, existing methods often manually assign the model importance, rendering them feasible only for LLMs with similar parameter alterations, such as multiple FT LLMs. The diverse parameter changed ranges between FT and PT LLMs pose challenges for current solutions in empirically determining the optimal combination. In this paper, we make a pioneering effort to broaden the applicability of merging techniques from FT to PT LLMs. We initially examine the efficacy of current methods in merging FT and PT LLMs, discovering that they struggle to deal with PT LLMs. Subsequently, we introduce an approach based on **WeIght DisENtanglement** (WIDEN) to effectively extend the merging scope, which first disentangles model weights into magnitude and direction components, and then performs adaptive fusion by considering their respective contributions. In the experiments, we merge Qwen1.5-Chat (an FT LLM with instruction-following skills) with Sailor (a PT LLM with multilingual abilities) across 1.8B, 4B, 7B, and 14B model sizes. Results reveal that: (1) existing solutions usually fail when merging Sailor, either losing both abilities or only retaining instruction-following skills; (2) WIDEN successfully injects the multilingual abilities of Sailor into Qwen1.5-Chat and make it proficient in Southeast Asian languages, achieving enhancements in the fundamental capabilities. In light of previous research, we also merge multiple 13B FT LLMs and observe that WIDEN achieves a balance of instruction following, mathematical reasoning, and code generation skills.

## 1 Introduction

In recent years, model merging has sparked significant interest as a prominent topic, which intends to integrate multiple homologous models (sharing the same backbone) into a singular one that encapsulates all the abilities (Wortsman et al., 2022; Matena & Raffel, 2022; Ilharco et al., 2023; Jin et al., 2023; Jang et al., 2024; Yadav et al., 2023; Davari & Belilovsky, 2023; Yu et al., 2024). Distinct from other approaches that can also amalgamate various skills (e.g., ensemble learning (Mohammed & Kora, 2023), multi-task learning (Crawshaw, 2020; Zhang & Yang, 2022)), model merging is lauded for its computational frugality, especially when applied to Large Language Models (LLMs). Notably, it achieves integration without using additional training data or even GPUs, establishing a new paradigm for efficiently combining LLMs' capabilities (Yu et al., 2024).

Technically, there are predominantly two strategies to equip LLMs with desired capabilities (Zhao et al., 2023): fine-tuning to elicit existing skills (Wang et al., 2023; Zhang et al., 2023a) and pre-training to inject new abilities (Wu et al., 2024). Existing merging methods mainly focus on integrating the skills of Fine-Tuned (FT) LLMs with minor parameter changes relative to the backbone, typically within 0.002 (Yu et al., 2024). However, it is crucial to acknowledge that pre-training is the cornerstone for fundamentally enhancing the capabilities of LLMs. The practicality of merging techniques in scenarios where Pre-Trained (PT) LLMs undergo substantial parameter shifts remains

unexplored, as depicted in Figure 1. Consequently, if the application of merging is restricted to FT LLMs, its potential for broader improvement would be significantly constrained.

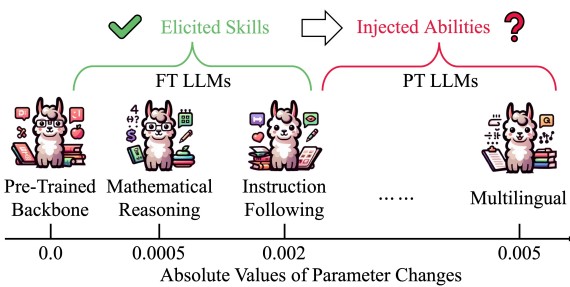

Figure 1: Issues of existing merging techniques.

Table 1: Average results of merging Qwen1.5-14B-Chat and Sailor-14B. Metrics of the best methods in Arithmetic, Geometric, and Pruning categories are reported.

|  | Instruction Following | Multilingual |
| --- | --- | --- |
| Qwen1.5-14B-Chat | 68.08 | 53.74 |
| Sailor-14B | 64.02 | 59.90 |
| Arithmetic-based | 66.30 (-1.78) | 40.72 (-19.18) |
| Geometric-based | 67.59 (-0.49) | 49.52 (-10.38) |
| Pruning-based | 51.72 (-16.36) | 28.69 (-31.21) |
| WIDEN | 66.75 (-1.33) | 59.67 (-0.23) |

To fill in the aforementioned blank, this work makes two key technical contributions.

**We examine the feasibility of existing approaches in absorbing the abilities from PT LLMs**. We investigate the performance of widely used arithmetic-based (Wortsman et al., 2022; Ilharco et al., 2023), geometric-based (Shoemake, 1985; Jang et al., 2024), and pruning-based (Yadav et al., 2023; Davari & Belilovsky, 2023; Yu et al., 2024) methods when merging FT and PT LLMs. As illustrated in Table 1, we find current methods either lose efficacy in retaining the abilities of PT LLMs (leading to a decrease of approximately 10 to 20 points on average) or fail to preserve both capabilities (resulting in an average degradation of about 15 and 30 points, respectively). One possible reason is that existing methods depend on manually assigned scaling terms to gauge the model contribution, which is only applicable when multiple LLMs depict comparable parameter alterations. Nonetheless, when confronted with diverse parameter changed ranges between FT and PT LLMs, deriving the optimal scaling factors according to human expertise becomes exceedingly arduous.

**We propose a new solution grounded in WeIght DisENtanglement (WIDEN) to expand the scope of merging techniques from FT to PT LLMs**. WIDEN tackles the drawbacks of existing works by automatically computing the model importance in the merging process without requiring manual specification, mitigating the influence induced by diverse parameter changed ranges between FT and PT LLMs. To be specific, WIDEN first disentangles each weight of a given LLM into two components: *magnitude* and *direction*. Then, the divergence of each component relative to the backbone is quantified to provide a numerical measure of how much each LLM has been altered. Next, WIDEN employs a ranking mechanism within each LLM to obtain the weight importance, tackling the diversity in parameter changed ranges between FT and PT LLMs. Finally, WIDEN performs adaptive merging on multiple LLMs by Softmax with the score calibration design.

We experiment with Qwen1.5-Chat (Bai et al., 2023) (an FT LLM with instruction-following skills) and Sailor (Dou et al., 2024) (a PT LLM with multilingual abilities for South-East Asia) across 1.8B, 4B, 7B, and 14B model scales to verify the effectiveness of WIDEN for model merging[1]. Experimental results indicate that WIDEN outperforms existing methods by not only absorbing the multilingual abilities of Sailor but also preserving the instruction-following skills of Qwen1.5-Chat. For example, in Table 1, WIDEN slightly causes an average reduction of 0.23 and 1.33 points for Sailor-14B and Qwen1.5-14B-Chat, respectively. These observations demonstrate that WIDEN effectively extends the applicability of merging techniques from FT to PT LLMs. Considering previous works, we further merge three FT LLMs including WizardLM-13B (Xu et al., 2024) for instruction following, WizardMath-13B (Luo et al., 2023) for mathematical reasoning, and llama-2-13b-code-alpaca (Chaudhary, 2023) for code generation. Results show that WIDEN is also feasible under the conventional setting and can strike a favorable balance among these capabilities.

Resources are available at `https://anonymous.4open.science/r/MergeLLM-5E0D`.

---

[1]To the best of our knowledge, Sailor is one of the few publicly accessible PT LLM that has undergone sufficient continued pre-training upon the open-source Qwen1.5 model (see Section A.6 and Section A.7 for more details), ideally suitable to our experimental scenarios. Therefore, Sailor and its homologous counterpart, Qwen1.5-Chat, are selected for our study.

## 2 RELATED WORK

**Fine-Tuning and Pre-Training of LLMs**. Generally, LLMs can be adapted to various tasks via two strategies: fine-tuning and pre-training (Zhao et al., 2023). Fine-tuning is designed to elicit backbones with specific skills by optimizing them on a limited set of task-specific data, obtaining FT LLMs with skills such as instruction following (Rafailov et al., 2023; Song et al., 2024) and mathematical reasoning (Yuan et al., 2023; Luo et al., 2023). The fine-tuning process typically brings minor modifications to the model parameters (Yu et al., 2024), holding true for both full fine-tuning approaches (Radford et al., 2018; Devlin et al., 2019) and parameter-efficient fine-tuning techniques (Houlsby et al., 2019; Li & Liang, 2021; Lester et al., 2021; Hu et al., 2022). In contrast to fine-tuning, pre-training trains LLMs on large-scale raw corpora to enhance models with domain knowledge (Ke et al., 2022; 2023; Cheng et al., 2024), deriving PT LLMs with fundamental abilities like finance analysis (Xie et al., 2023) and law assistance (Colombo et al., 2024b). Pre-training often leads to more obvious parameter shifts than fine-tuning due to extensive data used during the phase. Different from current merging methods that are only applicable to FT LLMs, this paper proposes a new solution to innovatively harness the capabilities of PT LLMs.

**Merging of LLMs**. Model merging aims to amalgamate multiple homologous models (derived from the same backbone) into a single one that possesses all the abilities (Wortsman et al., 2022; Matena & Raffel, 2022; Ilharco et al., 2023; Jin et al., 2023; Jang et al., 2024; Yadav et al., 2023; Davari & Belilovsky, 2023; Yu et al., 2024). The allure of the model merging technique stems from its minimal computational expense, particularly favorable for LLMs, which can be realized without retraining or GPUs (Yu et al., 2024). Existing merging techniques that are feasible for LLMs can be broadly categorized into three groups, which are based on arithmetic, geometric, and pruning. Average Merging (Wortsman et al., 2022) and Task Arithmetic (Ilharco et al., 2023) belong to arithmetic-based approaches. The former utilizes averaged parameters to create the merged model, whereas the latter introduces the concept of task vector (i.e., parameter difference between an FT model and its backbone) and uses a scaling term to regulate the importance of various models. As geometric-based methods, both SLERP (Shoemake, 1985) and Model Stock (Jang et al., 2024) consider the geometric properties in weight space. In particular, SLERP is specifically designed for the integration of two models, which performs spherical interpolation of model weights. Model Stock approximates a center-close weight based on several FT models, utilizing their backbone as an anchor point. TIES-Merging (Yadav et al., 2023), Breadcrumbs (Davari & Belilovsky, 2023), and DARE (Yu et al., 2024) are methods based on pruning. TIES-Merging eliminates parameter interference among multiple models by first removing delta parameters with low magnitudes and then merging parameters with consistent signs after resolving disagreements. Breadcrumbs masks out the extreme tails (also known as outliners) of the absolute magnitude distribution of task vectors to obtain the final model. DARE is a versatile plug-in for existing merging approaches, which first randomly drops delta parameters and then rescales the remaining ones to maintain model performance. However, most of the current methods manually determine the importance of each model, suitable only for LLMs with similar parameter changes. When the parameter changed ranges are diverse between FT and PT LLMs, determining the optimal combination becomes overwhelmingly challenging. This paper initially verifies the limitations of existing methods in combining the abilities of PT LLMs. Subsequently, an approach based on weight disentanglement is introduced to effectively expand the scope of merging techniques from FT to PT LLMs.

## 3 METHODOLOGY

### 3.1 PRELIMINARIES

**Merging Beyond FT LLMs**. Given a collection of $N$ homologous LLMs characterized by parameters $\{\Theta^1, \Theta^2, \cdots, \Theta^N\}$, all of which share the same backbone with parameters $\Theta_{\text{PRE}}$, model merging aims to amalgamate the parameters of $N$ LLMs into a singular model with all the capabilities, denoted as $\Theta_{\text{M}}$. Previous studies only focus on combining the skills of FT LLMs parameterized by $\{\Theta_{\text{FT}}^1, \Theta_{\text{FT}}^2, \cdots, \Theta_{\text{FT}}^N\}$, where each model exhibits slight parameter changes, usually within 0.002 (Yu et al., 2024). In this paper, we extend the scope of merging techniques from FT to PT LLMs, intending to absorb the abilities of PT LLMs. Therefore, the parameters targeted for merging become $\{\Theta_{\text{TYPE}_1}^1, \Theta_{\text{TYPE}_2}^2, \cdots, \Theta_{\text{TYPE}_N}^N\}$, where $\text{TYPE}_n$ ($1 \leq n \leq N$) can be either FT or PT.

**Weight Disentanglement**. As outlined in Salimans & Kingma (2016); Liu et al. (2024), a weight $W \in \mathbb{R}^{d \times k}$ can be disentangled into two components: a row vector $m \in \mathbb{R}^{1 \times k}$ that captures the magnitudes and a matrix $D \in \mathbb{R}^{d \times k}$ that stores the direction vectors. Here, $d$ and $k$ represent the output and input dimensions. Mathematically, the disentanglement of weight $W$ is achieved by

$$W = mD = \|W\|_c \frac{W}{\|W\|_c} \in \mathbb{R}^{d \times k}, \tag{1}$$

where $\| \cdot \|_c$ denotes the vector-wise $l_c$-norm of a matrix across each column. Such a decoupling operation guarantees that each column $D_{:,j}$ ($1 \le j \le k$) is a unit vector, and scalar $m_j \in m$ signifies the magnitude of direction vector $D_{:,j}$. Since the primary challenge of extending merging scope to PT LLMs lies in the manual assignment of model importance, we employ weight disentanglement to initially decouple weights into magnitudes and directions, and then automatically compute the weight importance without human expertise based on these two components.

## 3.2 EXPLORING EFFICACY OF CURRENT METHODS WHEN MERGING PT LLMS

We investigate the efficacy of seven commonly used merging techniques when integrating the abilities of PT LLMs. To be specific, Average Merging (Wortsman et al., 2022) and Task Arithmetic (Ilharco et al., 2023) are arithmetic-based methods. SLERP (Shoemake, 1985) and Model Stock (Jang et al., 2024) belong to geometric-based approaches. TIES-Merging (Yadav et al., 2023), Breadcrumbs (Davari & Belilovsky, 2023) and DARE (Yu et al., 2024) are pruning-based solutions. Please see Section A.4 for detailed descriptions of these methods. To evaluate the performance, we attempt to combine the instruction-following skills of an FT LLM, Qwen1.5-Chat (Bai et al., 2023), and the multilingual abilities of a PT LLM, Sailor (Dou et al., 2024). Experimental setup, results, and analysis can be found in Section 4.

Since this part mainly concentrates on the feasibility of merging techniques when applied to PT LLMs, we highlight the key conclusion pertinent to PT LLMs: *existing merging approaches face difficulties in preserving the abilities of PT LLMs*. As evidenced in Table 2, the performance of all merging methods on the multilingual abilities significantly declines. This phenomenon is largely attributed to the reliance of most methods on manually assigned scaling factors to determine the contribution of each model at various levels throughout the merging process, encompassing model level (Ilharco et al., 2023; Yadav et al., 2023; Davari & Belilovsky, 2023), layer/module level (Goddard et al., 2024), and parameter level (Shoemake, 1985). The diverse parameter changed ranges between FT and PT LLMs complicate the manual assignment of model importance, making it intractable to define optimal scaling factors case by case.

## 3.3 EXTENDING MERGING SCOPE TO PT LLMS VIA WEIGHT DISENTANGLEMENT

We present a new approach based on **WeIght DisENtanglement** (WIDEN) to innovatively broaden the applicability of model merging techniques from FT to PT LLMs, whose key concept is to adaptively assess the importance of weights during the merging process for neutralizing the effects of diverse parameter changed ranges between FT and PT LLMs. As shown in Figure 4 in Section A.1, WIDEN mainly comprises four steps. Given the weights of LLMs (including the backbone as well as models to be merged), WIDEN 1) disentangles each weight into a row vector of magnitudes and a matrix of direction vectors; 2) estimates weight divergence relative to the backbone founded on absolute values of magnitude alterations and cosine similarities between direction vectors; 3) ranks the weights inside each LLM grounded in their divergence to derive the weight importance, thereby mitigating the impact of diverse parameter changed ranges; 4) merges multiple LLMs into a single one according to the obtained weight importance via Softmax with score calibration.

**Disentangling Weights of LLMs**. Given multiple homologous LLMs (each LLM can be obtained by either FT or PT) with parameters $\{\Theta^1, \Theta^2, \cdots, \Theta^N\}$ as well as the backbone with parameters $\Theta_{\text{PRE}}$, we first perform weight disentanglement for the parameters. Take $W^n \in \Theta^n$ with shape $\mathbb{R}^{d \times k}$ as an example[2]. $W^n$ can be decoupled into $m^n = \|W^n\|_c \in \mathbb{R}^{1 \times k}$ and $D^n = \frac{W^n}{\|W^n\|_c} \in \mathbb{R}^{d \times k}$. After applying this disentanglement across all the LLMs, we can obtain the sets of row vectors of magnitudes $\{m^n\}_{n=1}^N \cup \{m_{\text{PRE}}\}$ and matrices of direction vectors $\{D^n\}_{n=1}^N \cup \{D_{\text{PRE}}\}$.

---

[2]Note that $\Theta^n$ represents the collection of parameters of the $n$-th LLM, consisting of a multitude of weights.

**Estimating Weight Divergence Relative to Backbone**. We estimate the weight divergence of each LLM relative to the backbone from the perspective of magnitudes and directions with two measurements. To be specific, we compute the absolute values of magnitude alterations and determine the changes between direction vectors based on cosine similarities as follows,

$$\Delta \boldsymbol{m}^n = |\boldsymbol{m}^n - \boldsymbol{m}_{\text{PRE}}| \in \mathbb{R}^{1 \times k}, \text{ for } 1 \leq n \leq N,$$
$$\Delta D_j^n = 1 - \text{CosineSimilarity}(\boldsymbol{D}_{:,j}^n, \boldsymbol{D}_{\text{PRE:},j}) \in \mathbb{R}, \text{ for } 1 \leq j \leq k, \ 1 \leq n \leq N, \tag{2}$$

where $\text{CosineSimilarity}(\boldsymbol{x}, \boldsymbol{y}) = \frac{\boldsymbol{x} \cdot \boldsymbol{y}}{\|\boldsymbol{x}\|_2 \cdot \|\boldsymbol{y}\|_2}$. Thus, we obtain the divergences of the LLMs relative to the backbone in both magnitudes $\{\Delta \boldsymbol{m}^n \in \mathbb{R}^{1 \times k}\}_{n=1}^N$ and directions $\{\Delta \boldsymbol{D}^n \in \mathbb{R}^{1 \times k}\}_{n=1}^N$.

**Ranking Weights Inside Each LLM**. We design a ranking mechanism to alleviate the potential impact of diverse parameter changed ranges among various LLMs, which assigns importance to the weights within each LLM according to their divergence relative to the backbone (greater divergence indicates higher essentiality). The ranking mechanism is applied to both the magnitudes and the directions of weights. To illustrate, consider the magnitudes as an instance. Given $\Delta \boldsymbol{m}^n \in \mathbb{R}^{1 \times k}$ of the $n$-th LLM, we initially sort $\Delta \boldsymbol{m}^n$ in ascending order, yielding an index row vector $\boldsymbol{m}_{\text{IND}}^n \in \mathbb{R}^{1 \times k}$ that contains values ranging from 1 to $k$. Subsequently, we derive a row vector $\widetilde{\boldsymbol{m}}^n \in \mathbb{R}^{1 \times k}$ that encapsulates normalized ranking scores based on $\boldsymbol{m}_{\text{IND}}^n$, which is computed by

$$\widetilde{m}_{m_{\text{IND}_j}^n}^n = j/k, \text{ for } 1 \leq j \leq k. \tag{3}$$

$\widetilde{\boldsymbol{m}}^n \in \mathbb{R}^{1 \times k}$ represents the normalized importance of each position within the range $[1, \cdots, k]$ for the $n$-th LLM. Following the same procedure, the directions of weights can also be assigned with normalized importance, which can be denoted by $\widetilde{\boldsymbol{D}}^n \in \mathbb{R}^{1 \times k}$. Such a ranking mechanism ensures that, within each LLM, the importance of magnitudes and directions is uniformly distributed between 0 and 1, thereby eliminating the potential influences arising from diverse parameter changed ranges between FT and PT LLMs. After applying the ranking operation for all the LLMs, we can ultimately obtain $\{\widetilde{\boldsymbol{m}}^n \in \mathbb{R}^{1 \times k}\}_{n=1}^N$ and $\{\widetilde{\boldsymbol{D}}^n \in \mathbb{R}^{1 \times k}\}_{n=1}^N$.

**Merging LLMs via Softmax with Score Calibration**. We employ an adaptive merging strategy for multiple LLMs through a Softmax function, complemented by score calibration. Initially, we calculate the importance scores for magnitudes and directions by applying the Softmax function to $\{\widetilde{\boldsymbol{m}}^n \in \mathbb{R}^{1 \times k}\}_{n=1}^N$ and $\{\widetilde{\boldsymbol{D}}^n \in \mathbb{R}^{1 \times k}\}_{n=1}^N$, yielding $\widetilde{\mathcal{M}}, \widetilde{\mathcal{D}} \in \mathbb{R}^{N \times k}$ by

$$\widetilde{\mathcal{M}}_{n,j} = \frac{\exp\left(\widetilde{m}_j^n\right)}{\sum_{n'=1}^N \exp\left(\widetilde{m}_j^{n'}\right)} \in \mathbb{R}, \ \widetilde{\mathcal{D}}_{n,j} = \frac{\exp\left(\widetilde{D}_j^n\right)}{\sum_{n'=1}^N \exp\left(\widetilde{D}_j^{n'}\right)} \in \mathbb{R}, \text{ for } 1 \leq j \leq k, \ 1 \leq n \leq N, \tag{4}$$

However, Softmax restricts the sum of parameter importance across multiple LLMs to 1, potentially diminishing the significance of crucial parameters in certain cases. Thus, we incorporate a score calibration operation to relax the constraint of Softmax for essential parameters. We identify crucial parameters as those whose importance exceeds the average level by a factor of $t$ as follows,

$$\mathbb{P}_m^n = \{j | \widetilde{m}_j^n > \frac{t}{k} \cdot \sum_{j'=1}^k \widetilde{m}_{j'}^n\}, \ \mathbb{P}_D^n = \{j | \widetilde{D}_j^n > \frac{t}{k} \cdot \sum_{j'=1}^k \widetilde{D}_{j'}^n\}. \tag{5}$$

Subsequently, we calibrate the scores using $\mathbb{P}_m^n$ and $\mathbb{P}_D^n$ by

$$\mathcal{M}_{n,j} = \begin{cases} s, & \text{if } j \in \mathbb{P}_m^n \\ \widetilde{\mathcal{M}}_{n,j}, & \text{if } j \notin \mathbb{P}_m^n \end{cases}, \ \mathcal{D}_{n,j} = \begin{cases} s, & \text{if } j \in \mathbb{P}_D^n \\ \widetilde{\mathcal{D}}_{n,j}, & \text{if } j \notin \mathbb{P}_D^n \end{cases}, \tag{6}$$

where $s$ regulates the numerical value of score calibration. Finally, we integrate the weights of multiple LLMs into $\boldsymbol{W}_{\text{M}}$ by considering the adjusted contributions of both magnitudes and directions,

$$\boldsymbol{W}_{\text{M}} = \boldsymbol{W}_{\text{PRE}} + \sum_{n=1}^N \frac{\mathcal{M}_{n,:} + \mathcal{D}_{n,:}}{2} \odot (\boldsymbol{W}^n - \boldsymbol{W}_{\text{PRE}}) \in \mathbb{R}^{d \times k}. \tag{7}$$

Note that $t$ and $s$ are designed to control the merging importance of parameters after applying the Softmax function. If more parameters are desired to be assigned with higher importance, $t$ should be reduced and $s$ should be increased. Conversely, $t$ should be increased and $s$ should be reduced.

*Remark 1.* The aforementioned procedure is designed to deal with two-dimensional weights within LLMs, accounting for both magnitudes and directions. For one-dimensional parameters, such as weights in normalization layers and biases in linear transformations, we handle them as vectors of magnitudes and estimate their changes relative to the backbone by absolute values of the differences.

*Remark 2.* Existing arithmetic-based merging methods including Average Merging (Wortsman et al., 2022) and Task Arithmetic (Ilharco et al., 2023), can be viewed as special instances of the proposed WIDEN. Specifically, the computation procedure of Average Merging (Wortsman et al., 2022) for $N$ LLMs is denoted by

$$\boldsymbol{W}_{\mathrm{M}} = \frac{1}{N}\sum_{n=1}^{N}\boldsymbol{W}^{n} = \boldsymbol{W}_{\mathrm{PRE}} + \frac{1}{N}\sum_{n=1}^{N}\left(\boldsymbol{W}^{n} - \boldsymbol{W}_{\mathrm{PRE}}\right) \in \mathbb{R}^{d\times k}. \tag{8}$$

Task Arithmetic (Ilharco et al., 2023) is implemented as follows,

$$\boldsymbol{W}_{\mathrm{M}} = \boldsymbol{W}_{\mathrm{PRE}} + \lambda\sum_{n=1}^{N}\left(\boldsymbol{W}^{n} - \boldsymbol{W}_{\mathrm{PRE}}\right) \in \mathbb{R}^{d\times k}, \tag{9}$$

where $\lambda$ denotes the scaling term. It is straightforward that in Equation (5), if $t$ is set to be minus, all the parameters can be considered crucial, with their importance scores calibrated to $s$. Thus, Equation (7) can be rewritten as

$$\boldsymbol{W}_{\mathrm{M}} = \boldsymbol{W}_{\mathrm{PRE}} + \sum_{n=1}^{N}\frac{s+s}{2}\left(\boldsymbol{W}^{n} - \boldsymbol{W}_{\mathrm{PRE}}\right) = \boldsymbol{W}_{\mathrm{PRE}} + s\sum_{n=1}^{N}\left(\boldsymbol{W}^{n} - \boldsymbol{W}_{\mathrm{PRE}}\right) \in \mathbb{R}^{d\times k}. \tag{10}$$

To this end, when $t < 0.0$ and $s = 1/N$, WIDEN transforms into Average Merging; when $t < 0.0$ and $s = \lambda$, WIDEN represents Task Arithmetic.

## 4 EXPERIMENTS

We conduct experiments on model merging in two scenarios: 1) integrating both FT and PT LLMs, a new setting not explored before; 2) combining FT LLMs as in previous research.

### 4.1 EXPERIMENTAL SETUP

**Merging FT and PT LLMs**. We choose Qwen1.5-Chat (Bai et al., 2023) with instruction-following skills as the FT LLM and select Sailor (Dou et al., 2024) with multilingual abilities for South-East Asia as the PT LLM. Both models adopt Qwen1.5 (Bai et al., 2023) as the backbone. Open LLM Leaderboard (Beeching et al., 2023) and benchmark for South-East Asian languages (Dou et al., 2024) are used for evaluating the performance of models across 1.8B, 4B, 7B, and 14B sizes.

**Merging FT LLMs**. In accordance with Yu et al. (2024), we merge three FT LLMs that are based on Llama-2-13b (Touvron et al., 2023): WizardLM-13B (Xu et al., 2024) for instruction following, WizardMath-13B (Luo et al., 2023) for mathematical reasoning, and llama-2-13b-code-alpaca (Chaudhary, 2023) for code generation. AlpacaEval 2.0 (Dubois et al., 2024), GSM8K (Cobbe et al., 2021), MATH (Hendrycks et al., 2021b), HumanEval (Chen et al., 2021), and MBPP (Austin et al., 2021) are utilized for evaluation.

Please see Section A.3 for the overview and evaluation metrics of the benchmarks. Also, refer to Table 7 in Section A.2 for the details of FT and PT LLMs. We compare WIDEN with seven popular baselines for model merging, including Average Merging (Wortsman et al., 2022), Task Arithmetic (Ilharco et al., 2023), SLERP (Shoemake, 1985), Model Stock (Jang et al., 2024), TIES-Merging (Yadav et al., 2023), Breadcrumbs (Davari & Belilovsky, 2023), and DARE (Yu et al., 2024). See Section 3.2 and Section A.4 for more descriptions.

**Configurations of Merging Methods**. We apply grid search to identify the optimal settings for various merging techniques. The proposed WIDEN utilizes $l_2$ normalization and involves two hyperparameters: $s$ and $t$. For ease of implementation, the score calibration factor $s$ is consistently fixed to 1.0 across all the cases. The factor $t$ is determined by grid search. Please refer to Table 8 in Section A.5 for detailed information about the searched ranges.

**Hardware Requirements**. The process of merging LLMs requires only CPU resources. To evaluate the merged LLMs, we employ A100 GPUs equipped with 80 GB of memory. Notably, all the experiments can be successfully reproduced using a single A100 GPU.

## 4.2 Performance of Merging FT and PT LLMs

Table 2 and Table 11 show the results of merging Qwen1.5-Chat and Sailor on South-East Asian language benchmark. Since Average Merging is a special case of Task Arithmetic when the scaling term is 0.5, we thereby only report the results of Task Arithmetic, which inherently include the performance of Average Merging. Note that th, id, vi, and jv are abbreviations of Thai, Indonesian, Vietnamese, and Javanese. The best and second-best results are marked in **bold** and underlined fonts. From Table 2, two conclusions can be summarized.

Table 2: Results of merging Qwen1.5-Chat and Sailor on South-East Asian language benchmark.

| Size | Models | Merging Methods | XQuAD th | TydiQA id | XQuAD vi | XCOPA th | id | vi | Belebele th | id | vi | M3Exam jv | Average | Average Rank |
|---|---|---|---|---|---|---|---|---|---|---|---|---|---|---|
| 7B | Qwen1.5 | / | 53.79/69.30 | 57.17/77.28 | 56.63/76.99 | 54.20 | 62.20 | 66.20 | 38.33 | 42.00 | 42.89 | 26.15 | 55.63 | / |
|  | Qwen1.5-Chat | / | 24.28/46.77 | 42.30/67.57 | 45.51/69.91 | 56.20 | 66.80 | 70.40 | 38.67 | 43.11 | 47.11 | 28.30 | 49.76 | / |
|  | Sailor | / | 57.88/71.06 | 60.53/75.42 | 53.81/74.62 | 59.00 | 72.20 | 72.20 | 41.56 | 44.33 | 45.33 | 32.88 | 58.52 | / |
|  | Qwen1.5-Chat & Sailor | Task Arithmetic | 28.20/49.62 | **45.84**/65.78 | 37.38/61.53 | **63.20** | **77.60** | **73.40** | 38.89 | 46.89 | 45.11 | 30.46 | 51.07 | 2.15 |
|  |  | SLERP | 16.62/43.62 | 20.53/54.02 | 33.70/61.49 | 55.80 | 73.40 | 73.00 | 38.44 | 47.89 | 47.56 | 28.30 | 45.72 | 3.23 |
|  |  | Model Stock | 26.72/52.69 | 24.78/58.88 | 43.80/69.50 | 54.60 | 66.00 | 69.40 | 37.33 | 42.78 | 43.67 | 27.76 | 47.53 | 3.31 |
|  |  | TIES-Merging | 0.61/8.84 | 5.66/17.23 | 7.70/20.78 | 50.20 | 62.20 | 59.80 | 30.22 | 35.33 | 35.11 | 25.07 | 27.60 | 5.54 |
|  |  | Breadcrumbs | 6.79/11.38 | 7.61/15.23 | 12.32/27.90 | 51.40 | 66.40 | 57.20 | 31.33 | 34.00 | 32.56 | 24.53 | 29.13 | 5.23 |
|  |  | WIDEN | **42.65/64.21** | **45.84/73.37** | **48.42/73.17** | 60.20 | 77.40 | **73.60** | **40.11** | **51.11** | **48.56** | **32.88** | 56.27 | **1.15** |
| 14B | Qwen1.5 | / | 55.53/74.39 | 60.35/81.07 | 57.66/77.62 | 58.40 | 70.40 | 72.60 | 41.22 | 48.67 | 44.44 | 26.15 | 59.12 | / |
|  | Qwen1.5-Chat | / | 33.59/59.98 | 37.17/65.46 | 44.14/71.91 | 61.80 | 75.20 | 71.80 | 44.00 | 51.00 | 52.67 | 29.92 | 53.74 | / |
|  | Sailor | / | 49.43/70.01 | 58.94/77.85 | 57.74/77.34 | 62.60 | 77.60 | 78.60 | 40.89 | 47.67 | 47.11 | 32.88 | 59.90 | / |
|  | Qwen1.5-Chat & Sailor | Task Arithmetic | 8.53/24.39 | 13.45/33.54 | 13.52/25.75 | 59.80 | **82.40** | **78.20** | **46.00** | **56.33** | **53.78** | **33.69** | 40.72 | 2.54 |
|  |  | SLERP | 14.53/44.70 | 22.48/61.67 | 42.69/69.48 | **61.80** | 75.60 | 74.60 | 43.22 | 52.56 | 50.56 | 29.92 | 49.52 | 2.46 |
|  |  | Model Stock | 25.59/53.10 | 14.87/51.19 | 44.74/70.20 | 58.60 | 70.40 | 71.80 | 42.67 | 49.89 | 45.11 | 27.22 | 48.11 | 3.08 |
|  |  | TIES-Merging | 0.44/8.78 | 1.42/12.87 | 0.00/6.95 | 55.20 | 69.20 | 67.20 | 32.78 | 39.00 | 37.11 | 27.22 | 27.55 | 5.46 |
|  |  | Breadcrumbs | 1.22/6.48 | 2.30/20.88 | 3.17/14.46 | 52.20 | 64.60 | 63.40 | 34.78 | 42.11 | 40.67 | 26.68 | 28.69 | 5.23 |
|  |  | WIDEN | **49.61/73.16** | **50.62/75.09** | **54.75/78.23** | 60.80 | 77.40 | 74.60 | 42.22 | 56.22 | 50.44 | 32.61 | **59.67** | **1.77** |

Firstly, *existing model merging approaches encounter significant challenges when incorporating the multilingual abilities of Sailor, leading to a marked decline in performance*. The downturn is probably attributed to the difficulty in determining the optimal combination due to diverse parameter changed ranges between Qwen1.5-Chat and Sailor. We also notice that the reduction is particularly pronounced in pruning-based methods, prompting us to conduct additional verifications. As demonstrated in Table 3, we find that the feasibility of pruning strategies such as DARE and Magnitude-based Pruning (MP) in TIES-Merging and Breadcrumbs is severely compromised with minor parameter drop rates on Sailor-7B, far below the levels reported results in the original studies (i.e., 0.9 in DARE, 0.8 in TIES-Merging, and 0.85 in Breadcrumbs), diminishing the effectiveness of pruning in alleviating parameter interference. As a result, DARE fails to serve as a plug-in for existing merging techniques when considering PT LLMs, and its inferior results are excluded.

Table 3: Performance of pruning strategies on Sailor-7B for Vietnamese-related tasks.

| | Drop Rate | XQuAD | XCOPA | Belebele |
|---|---|---|---|---|
| Sailor-7B | / | 53.81/74.62 | 72.20 | 45.33 |
| DARE | 0.1 | 47.56/66.95 | 64.20 | 41.00 |
|  | 0.3 | 5.90/16.05 | 55.60 | 30.56 |
| MP | 0.1 | 54.23/75.16 | 72.80 | 45.44 |
|  | 0.3 | 52.44/73.53 | 72.20 | 44.78 |
|  | 0.5 | 49.19/70.11 | 70.00 | 43.67 |
|  | 0.8 | 13.77/30.13 | 59.00 | 34.56 |

Secondly, *WIDEN effectively assimilates the multilingual capabilities of Sailor, emerging as the top performer among all the merging techniques*. The key advantage of WIDEN lies in the adaptive computation of weight importance by considering both magnitudes and directions during the merging process, mitigating the effects of diverse parameter changed ranges between FT and PT LLMs.

Table 4 and Table 12 depict the merging performance on Open LLM Leaderboard. We find that geometric-based approaches (SLERP and Model Stock) excel in retraining the instruction-following

Table 4: Performance of merging Qwen1.5-Chat and Sailor on Open LLM Leaderboard.

| Size | Models | Merging Methods | ARC | Hella-Swag | MMLU | Truthful-QA | Wino-grande | GSM8K | Average | Average Rank |
|------|--------|-----------------|-----|------------|------|-------------|-------------|-------|---------|--------------|
| | Qwen1.5 | / | 54.86 | 78.45 | 60.60 | 51.09 | 71.03 | 56.79 | 62.14 | / |
| | Qwen1.5-Chat | / | 56.14 | 78.71 | 60.18 | 53.61 | 67.48 | 54.21 | 61.72 | / |
| | Sailor | / | 49.57 | 76.13 | 52.91 | 40.07 | 71.35 | 34.65 | 54.11 | / |
| 7B | | Task Arithmetic | 52.05 | 75.15 | 59.38 | 50.84 | 69.77 | 25.55 | 55.46 | 3.50 |
| | Qwen1.5-Chat | SLERP | 54.78 | 76.20 | 60.76 | 50.78 | 71.51 | 55.50 | **61.59** | 2.33 |
| | & Sailor | Model Stock | **55.12** | **76.29** | **61.18** | 49.33 | 71.43 | **55.80** | 61.53 | **2.00** |
| | | TIES-Merging | 43.86 | 56.88 | 52.39 | 46.59 | 67.56 | 0.00 | 44.55 | 5.67 |
| | | Breadcrumbs | 47.18 | 49.99 | 52.66 | **52.05** | 64.88 | 0.45 | 44.53 | 4.67 |
| | | WIDEN | 53.84 | 76.25 | 57.65 | 49.34 | **71.90** | 44.81 | 58.97 | 2.83 |
| | Qwen1.5 | / | 56.40 | 81.22 | 67.79 | 52.04 | 74.43 | 68.01 | 66.65 | / |
| | Qwen1.5-Chat | / | 57.25 | 82.56 | 67.48 | 60.42 | 72.69 | 68.08 | 68.08 | / |
| | Sailor | / | 55.46 | 80.31 | 62.95 | 46.64 | 76.80 | 61.94 | 64.02 | / |
| 14B | | Task Arithmetic | 56.57 | **81.59** | 67.52 | **62.93** | 75.22 | 53.98 | 66.30 | 2.50 |
| | Qwen1.5-Chat | SLERP | 55.72 | 79.94 | 67.94 | 57.51 | 75.14 | **69.29** | **67.59** | 3.00 |
| | & Sailor | Model Stock | 57.00 | 80.50 | **68.44** | 51.98 | 76.01 | 66.72 | 66.77 | **2.33** |
| | | TIES-Merging | 49.74 | 67.23 | 60.54 | 47.43 | 72.14 | 0.30 | 49.56 | 5.67 |
| | | Breadcrumbs | 51.88 | 62.22 | 63.47 | 57.90 | 70.32 | 4.55 | 51.72 | 4.83 |
| | | WIDEN | **57.17** | 80.05 | 66.00 | 54.85 | **76.09** | 66.34 | 66.75 | 2.67 |

skills of Qwen1.5-Chat, indicating that parameters of FT LLMs may potentially exhibit more evident properties in the geometric space. WIDEN shows competitive results alongside SLERP and Model Stock, underscoring its applicability in merging FT LLMs. Moreover, WIDEN outperforms arithmetic-based methods since it is a generalized format of these methods and offers greater flexibility through the adaptive computation of weight importance. The performance of WIDEN consistently improves with increasing model sizes, indicating its potential scalability. Although WIDEN achieves competitive but not state-of-the-art performance on the Open LLM Leaderboard, it consistently delivers satisfactory results across both benchmarks, while most baselines fail to do so, demonstrating the robustness and generalizability of WIDEN.

### 4.3 PERFORMANCE OF MERGING FT LLMs

Under the setting of merging multiple FT LLMs, we strictly follow the identical protocol in Yu et al. (2024) and report the official results in Table 5 for fair comparisons. One exception is that we use AlpacaEval 2.0 instead of AlpacaEval in Yu et al. (2024) for evaluation, aiming to provide more convincing and reliable verifications. Since SLERP is only applicable for dealing with two models, its results for merging three LLMs are unavailable.

From Table 5, we observe that the efficacy of certain baselines drastically fluctuates when integrating FT LLMs. For example, Model Stock appears to lose potency, whereas pruning-based methods including TIES-Merging and Breadcrumbs show competitive performance. WIDEN consistently depicts results that are on par with established merging techniques in most situations, affirming its suitability in the standard setting of merging multiple FT LLMs. It is worth noting that WIDEN performs competitively but less prominently than baselines when merging multiple FT models. This is because WIDEN excels at merging LLMs with obvious differences in parameter changed ranges by disentangling parameters into magnitudes and directions. In the case of FT models with minor and similar parameter changes, treating weights holistically or disentangling them leads to minimal disparity, which makes the disentanglement operation less pronounced.

### 4.4 INVESTIGATIONS OF DESIGNS IN WIDEN

The foundational designs in WIDEN consist of three components: weight disentanglement, ranking weights inside each model, and score calibration for Softmax. To assess the contribution of each module, we respectively remove the above components and measure the performance of the remaining parts. Specifically, we eliminate the disentanglement of weights by calculating the discrepancy between the weights of LLM and the corresponding backbone using cosine similarities, denoted as WIDEN w/o WD. We substitute the ranking mechanism with min-max normalization within each model, represented by WIDEN w/o RANK. We discard the score calibration and directly employ

Table 5: Performance of merging WizardLM-13B, WizardMath-13B, and llama-2-13b-code-alpaca.

| Models | Merging Methods | Instruction-following | Mathematical Reasoning | | Code Generation | |
|---|---|---|---|---|---|---|
| | | AlpacaEval 2.0 | GSM8K | MATH | HumanEval | MBPP |
| WizardLM-13B | / | 12.73 | 2.20 | 0.04 | 36.59 | 34.00 |
| WizardMath-13B | / | / | 64.22 | 14.02 | / | / |
| llama-2-13b-code-alpaca | / | / | / | / | 23.78 | 27.60 |
| WizardLM-13B & WizardMath-13B | Task Arithmetic | **11.85** | **66.34** | 13.40 | 28.66 | 30.60 |
| | SLERP | 7.90 | 66.19 | 13.44 | 28.05 | 30.80 |
| | Model Stock | 0.25 | 0.00 | 0.00 | 3.05 | 25.80 |
| | TIES-Merging | 10.07 | 15.77 | 2.04 | **37.80** | **35.60** |
| | Breadcrumbs | 9.85 | 64.75 | 11.80 | 26.22 | 33.20 |
| | WIDEN | 9.45 | **66.34** | **13.58** | 28.66 | 30.40 |
| WizardLM-13B & llama-2-13b-code-alpaca | Task Arithmetic | **10.09** | / | / | 31.70 | 32.40 |
| | SLERP | 6.04 | / | / | 32.32 | **35.80** |
| | Model Stock | 0.25 | / | / | 3.66 | 24.80 |
| | TIES-Merging | 7.27 | / | / | 0.00 | 0.00 |
| | Breadcrumbs | 7.23 | / | / | **33.54** | 32.00 |
| | WIDEN | 6.53 | / | / | 31.70 | 35.60 |
| WizardMath-13B & llama-2-13b-code-alpaca | Task Arithmetic | / | **64.67** | **13.98** | 8.54 | 8.60 |
| | SLERP | / | 61.41 | 12.50 | 9.15 | 22.40 |
| | Model Stock | / | 0.00 | 0.00 | 4.27 | **25.60** |
| | TIES-Merging | / | 63.23 | 13.56 | **9.76** | 22.40 |
| | Breadcrumbs | / | 62.55 | 12.48 | 9.15 | 16.20 |
| | WIDEN | / | 64.22 | 13.58 | **9.76** | 9.80 |
| WizardLM-13B & WizardMath-13B & llama-2-13b-code-alpaca | Task Arithmetic | **11.51** | 58.45 | 9.88 | 18.29 | 29.80 |
| | Model Stock | 0.12 | 0.00 | 0.00 | 5.49 | 23.40 |
| | TIES-Merging | 9.22 | **62.55** | 9.54 | 21.95 | 30.40 |
| | Breadcrumbs | 10.89 | **62.55** | **10.58** | **23.78** | 29.60 |
| | WIDEN | 8.71 | 57.16 | 9.60 | 22.56 | **30.80** |

Softmax to compute importance scores, identified as WIDEN w/o SC. Figure 2 shows the impact of these three modifications, where OLL and SEA are the abbreviations for Open LLM Leaderboard and South-East Asian language benchmark, respectively. Note that the reported results are the average of metrics across all the datasets within each benchmark.

From Figure 2, we find that each design in WIDEN contributes to enhancing the merging performance, particularly in absorbing the multilingual abilities on the South-East Asian language benchmark. Precisely, the weight disentanglement refines the estimation of weight importance at a granular level, considering both magnitude and direction. The ranking mechanism offers a smoother distribution of weight importance based on continuous indices, effectively mitigating the influence of diverse parameter changed ranges. The calibration of scores computed by Softmax reallocates importance to critical parameters, which maintains the characteristics of essential parameters across multiple models. In summary, the components of WIDEN are indispensable and improve performance with varied benefits; the removal of any module leads to diminished outcomes.

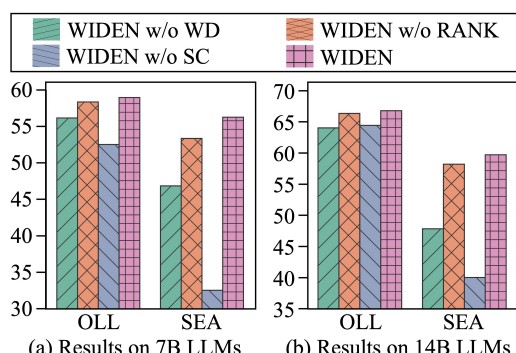

Figure 2: Effects of various designs in WIDEN.

### 4.5 ANALYSIS OF COMPUTED WEIGHT IMPORTANCE

We further delve into the properties of weight importance calculated by WIDEN from both qualitative and quantitative perspectives. Since Figure 2 demonstrates that the improvements in weight disentanglement and score calibration are notably more pronounced, we qualitatively depict the distribution of weight importance computed by WIDEN, WIDEN w/o WD, and WIDEN w/o SC on 7B model size in Figure 3. Our observations reveal that: 1) WIDEN exhibits a more balanced and

reasonable weight importance distribution than WIDEN w/o WD, attributed to the disentanglement of weights. The distribution of WIDEN ranges approximately from 0.3 to 0.8 and 0.9 to 1.0, versus 0.3 to 0.6 and 0.9 to 1.0 for WIDEN w/o WD. WIDEN considers the collective contributions of magnitude and direction, rather than the individual impacts of weights, leading to a more holistic assessment of weight importance with increased numbers of weights falling within the importance range from 0.6 to 0.8. As a result, compared with WIDEN w/o WD, WIDEN achieves 4.98% and 20.08% improvements on average on the Open LLM Leaderboard and the South-East Asian language benchmark, respectively; 2) In contrast to WIDEN w/o SC, WIDEN distinguishes essential weights and assigns high importance within the range of 0.6 to 0.8 as well as 0.9 to 1.0 for certain weights, thanks to the design of score calibration. Therefore, WIDEN ensures the retention of essential weights in both Qwen1.5-7B-Chat and Sailor-7B, resulting in 12.25% and 72.87% average enhancements on the two benchmarks.

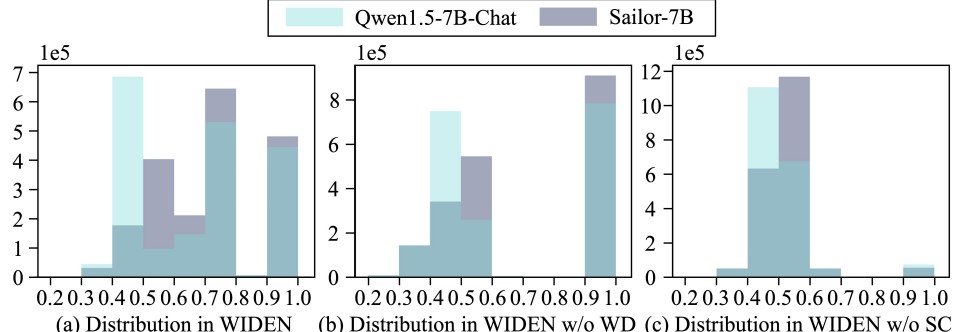

Figure 3: Distribution of weight importance computed by WIDEN and its variations.

Furthermore, we categorize weight importance into three levels: Low (L), Medium (M), and High (H). The Low tier comprises the first third of weights when sorted by ascending importance, indicating those with the least significance. The Medium tier includes weights from the 1/3 mark to the 2/3 mark, and the High tier contains weights from the 2/3 mark to the end. Table 6 quantitatively illustrates the adjustments of weight importance made by WIDEN when compared to WIDEN w/o WD and WIDEN w/o SC across three levels. We find that WIDEN effectively reallocates the weight importance via three aspects: 1) elevating weights of lower importance from Low to Medium; 2) either demoting or promoting weights of medium importance from Medium to Low or from Medium to High, respectively; 3) decreasing weights of high importance from High to Medium. These adjustments in weight importance explain how WIDEN brings improvements through the designs of weight disentanglement and score calibration.

Table 6: Adjustments of weight importance made by WIDEN.

| Adjustments | Models | L→L | L→M | L→H | M→L | M→M | M→H | H→L | H→M | H→H |
|---|---|---|---|---|---|---|---|---|---|---|
| WIDEN w/o WD | Qwen1.5-7B-Chat | 18.82% | 11.09% | 3.42% | 13.97% | 10.18% | 9.18% | 0.54% | 12.06% | 20.75% |
| to WIDEN | Sailor-7B | 15.34% | 10.50% | 7.48% | 17.80% | 7.72% | 7.80% | 0.18% | 15.10% | 18.07% |
| WIDEN w/o SC | Qwen1.5-7B-Chat | 24.78% | 7.69% | 0.85% | 7.93% | 17.51% | 7.88% | 0.62% | 8.12% | 24.61% |
| to WIDEN | Sailor-7B | 22.01% | 9.52% | 1.80% | 9.63% | 15.14% | 8.56% | 1.69% | 8.67% | 22.99% |

## 5 CONCLUSION

In this study, we paved the way for extending the merging scope from FT to PT LLMs. Specifically, we first observed that existing methods struggled to integrate the abilities of PT LLMs and then introduced WIDEN, an innovative approach based on weight disentanglement, to effectively deploy merging strategies to PT LLMs. Experimental findings demonstrated that WIDEN not only exhibited an advantage in absorbing the abilities of PT LLMs but also preserved the skills of FT LLMs. Additionally, WIDEN achieved competitive performance with established merging methods in the conventional setting of merging FT LLMs. We further offered a detailed analysis of the designs underlying WIDEN. This work made the first attempt to broaden the sources of combinable abilities, fostering the broader application of model merging techniques.

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

# A APPENDIX

## A.1 COMPUTATION PROCESS OF WIDEN

Figure 4 illustrates the framework of WIDEN.

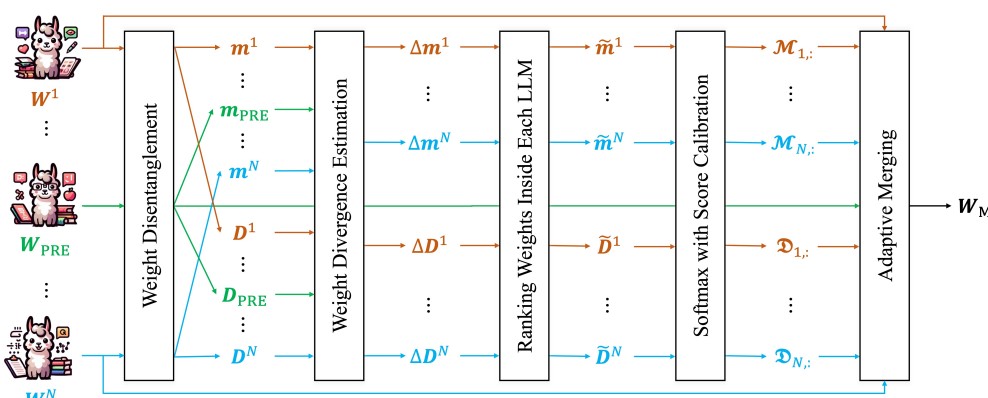

Figure 4: Framework of the proposed WIDEN.

## A.2 DETAILS OF FT AND PT LLMS

Table 7 depicts the versions and correspondences with backbones of FT and PT LLMs.

Table 7: Versions and correspondences with backbones of FT and PT LLMs.

| Types | Models | Backbones |
|---|---|---|
| FT LLM | Qwen1.5-1.8B-Chat[3] | Qwen1.5-1.8B[4] |
| PT LLM | Sailor-1.8B[5] | Qwen1.5-1.8B[4] |
| FT LLM | Qwen1.5-4B-Chat[6] | Qwen1.5-4B[7] |
| PT LLM | Sailor-4B[8] | Qwen1.5-4B[7] |
| FT LLM | Qwen1.5-7B-Chat[9] | Qwen1.5-7B[10] |
| PT LLM | Sailor-7B[11] | Qwen1.5-7B[10] |
| FT LLM | Qwen1.5-14B-Chat[12] | Qwen1.5-14B[13] |
| PT LLM | Sailor-14B[14] | Qwen1.5-14B[13] |
| FT LLM | WizardLM-13B[15] | Llama-2-13b[16] |
| | WizardMath-13B[17] | Llama-2-13b[16] |
| | llama-2-13b-code-alpaca[18] | Llama-2-13b[16] |

---

[3] https://huggingface.co/Qwen/Qwen1.5-1.8B-Chat
[4] https://huggingface.co/Qwen/Qwen1.5-1.8B
[5] https://huggingface.co/sail/Sailor-1.8B
[6] https://huggingface.co/Qwen/Qwen1.5-4B-Chat
[7] https://huggingface.co/Qwen/Qwen1.5-4B
[8] https://huggingface.co/sail/Sailor-4B
[9] https://huggingface.co/Qwen/Qwen1.5-7B-Chat
[10] https://huggingface.co/Qwen/Qwen1.5-7B
[11] https://huggingface.co/sail/Sailor-7B
[12] https://huggingface.co/Qwen/Qwen1.5-14B-Chat
[13] https://huggingface.co/Qwen/Qwen1.5-14B
[14] https://huggingface.co/sail/Sailor-14B
[15] https://huggingface.co/WizardLM/WizardLM-13B-V1.2
[16] https://huggingface.co/meta-llama/Llama-2-13b-hf
[17] https://huggingface.co/WizardLM/WizardMath-13B-V1.0
[18] https://huggingface.co/layoric/llama-2-13b-code-alpaca

## A.3 OVERVIEW AND EVALUATION METRICS OF BENCHMARKS

The Open LLM Leaderboard is established to assess open-source LLMs using the Eleuther AI Language Model Evaluation Harness (Gao et al., 2023), which encompasses six datasets: AI2 Reasoning Challenge (ARC) (Clark et al., 2018), HellaSwag (Zellers et al., 2019), MMLU (Hendrycks et al., 2021a), TruthfulQA (Lin et al., 2022), Winogrande (Sakaguchi et al., 2020), and GSM8K (Cobbe et al., 2021). These datasets adopt accuracy as the evaluation metric under various shot settings (25-, 10-, 0-, 5-, 5-, and 5-shot, respectively). The leaderboard ranks models based on the average scores across these six datasets.

The benchmark for South-East Asian languages is designed with four tasks: XQuAD (Artetxe et al., 2020) (Thai, Vietnamese) and TydiQA (Clark et al., 2020) (Indonesian) for question answering; XCOPA (Ponti et al., 2020) (Indonesian, Thai, Vietnamese) for commonsense reasoning; BELE-BELE (Bandarkar et al., 2023) (Indonesian, Thai, and Vietnamese) for reading comprehension; and M3Exam (Zhang et al., 2023b) (Javanese) for examination. All the datasets utilize 3-shot Exact Match (EM) and F1 as evaluation metrics. It is worth noticing that the official code[19] of Sailor computes multiple metrics for M3Exam on Thai and Vietnamese, which are inconsistent with the originally reported results. Thus, we only present the results of M3Exam (Javanese) in this work.

AlpacaEval 2.0 employs the win rate for assessment, calculated as the proportion of cases where a powerful LLM (GPT-4 Turbo is used in this work) prefers the outputs from the target model over those from GPT-4 Turbo. GSM8K and MATH are evaluated by zero-shot accuracy in addressing mathematical problems. HumanEval and MBPP adopt pass@1 as the evaluation metric, representing the fraction of individually generated code samples that successfully pass the unit tests.

## A.4 DESCRIPTIONS OF MODEL MERGING BASELINES

We compare with seven commonly-used model merging methods in the experiments:

- **Average Merging** simply averages the parameters of multiple models for building the merged model (Wortsman et al., 2022).
- **Task Arithmetic** employs a scaling term to modulate the importance of the backbone and various models to be merged (Ilharco et al., 2023).
- **SLERP** is tailored for the combination of two models, utilizing spherical interpolation to merge the model weights (Shoemake, 1985).
- **Model Stock** seeks to approximate a center-close weight by considering several FT models, where the backbone is leveraged as an anchor point (Jang et al., 2024).
- **TIES-Merging** aims to mitigate task conflicts in model merging by initially pruning delta parameters with lower magnitudes and subsequently fusing parameters that exhibit consistent signs (Yadav et al., 2023).
- **Breadcrumbs** refines model parameters by filtering out the extreme tails (i.e., outliers) in the absolute magnitude distribution of task vectors to derive the final merged model (Davari & Belilovsky, 2023).
- **DARE** serves as a versatile module for current merging techniques, which first randomly discards delta parameters and then rescales the remaining parameters to preserve the model performance (Yu et al., 2024).

## A.5 DETAILS OF GRID SEARCH ON HYPERPARAMETERS OF MERGING METHODS

Table 8 presents the searched ranges of hyperparameters of model merging approaches. We sample 10% of the data from each dataset in the benchmarks as the validation set for grid search. The settings that yield the best average performance on the validation set are selected for evaluation. This process is uniformly applied to all baseline methods as well as WIDEN to ensure a fair comparison.

For baselines like Task Arithmetic that rely on scaling terms, we select the optimal setting at the dataset level within the range [0.5, 1.0], rather than using an identical setting at the model level. We

---

[19] https://github.com/sail-sg/sailor-llm

find that on the Open LLM Leaderboard, Task Arithmetic performs better with a scaling term of 0.5 on some datasets and 1.0 on others. On the South-East Asian language benchmark, a scaling term of 1.0 consistently outperforms 0.5. For WIDEN, we aim to compute the importance of weights through weight disentanglement, eliminating the need for manual specification. Even for hyperparameters $t$ and $s$, we used a unified setting across all benchmarks. Such an implementation may reduce the advantage of WIDEN on the Open LLM Leaderboard to some extent but demonstrates its robustness and generalizability.

Table 8: Hyperparameter searched ranges of model merging approaches.

| Model Merging Methods | Search Ranges of Hyperparameters |
|---|---|
| Task Arithmetic | scaling term to merge parameters: [0.5, 1.0] |
| SLERP | spherical interpolation factor: [0.3, 0.5, 0.7] |
| Model Stock | / |
| TIES-Merging | scaling term to merge parameters: [0.5, 1.0], ratio to retain parameters with largest-magnitude values: [0.5, 0.7, 0.9] |
| Breadcrumbs | scaling term to merge parameters: [0.5, 1.0], ratio to mask parameters with largest-magnitude values: [0.01, 0.05], ratio to retain parameters [0.9] |
| WIDEN | factor to indicate the multiple above the average: [1.0, 2.0], factor to calibrate scores: [1.0] |

## A.6   ISSUES OF SEVERAL EXISTING PT LLMS

We present the statistics of some existing PT LLMs, including Sailor, finance-chat (Cheng et al., 2024), medicine-chat (Cheng et al., 2024), law-chat (Cheng et al., 2024), BioMistral-7B (Labrak et al., 2024), and Saul-7B-Base (Colombo et al., 2024a). Table 9 shows the information on domains and the number of training tokens of these PT LLMs.

Table 9: Domains and training tokens of some existing PT LLMs.

| Models | Backbones | Domains | Training Tokens |
|---|---|---|---|
| Sailor-1.8B[5] | Qwen1.5-1.8B[4] | Multilingual | 200B |
| Sailor-4B[8] | Qwen1.5-4B[7] | Multilingual | 200B |
| Sailor-7B[11] | Qwen1.5-7B[10] | Multilingual | 200B |
| Sailor-14B[14] | Qwen1.5-14B[13] | Multilingual | 200B |
| finance-chat[20] | Llama-2-7b-chat[21] | Finance Analysis | 1.2B |
| medicine-chat[22] | Llama-2-7b-chat[21] | Medical Analysis | 5.4B |
| law-chat[23] | Llama-2-7b-chat[21] | Law Assistance | 16.7B |
| BioMistral-7B[24] | Mistral-7B-Instruct-v0.1[25] | Medical Analysis | 3B |
| Saul-7B-Base[26] | Mistral-7B-v0.1[27] | Law Assistance | 30B |

It could be concluded that most current PT LLMs (except for Sailor) are pre-trained on fewer than 30B tokens, resulting in relatively small parameter changed ranges (see Table 10). This makes them less suitable for our experimental setup, as substantial parameter changes among the models to be merged are desired.

---

[20]https://huggingface.co/AdaptLLM/finance-chat
[21]https://huggingface.co/meta-llama/Llama-2-7b-chat-hf
[22]https://huggingface.co/AdaptLLM/medicine-chat
[23]https://huggingface.co/AdaptLLM/law-chat
[24]https://huggingface.co/BioMistral/BioMistral-7B
[25]https://huggingface.co/mistralai/Mistral-7B-Instruct-v0.1
[26]https://huggingface.co/Equall/Saul-7B-Base
[27]https://huggingface.co/mistralai/Mistral-7B-v0.1

## A.7 PARAMETER CHANGED RANGES OF FT AND PT LLMS

We depict the statistics about the deciles of parameter changed ranges of both FT and PT LLMs in Table 10, which are derived by first sorting the entire ranges and then indexing at positions corresponding to 0%, 10%, 20%, ..., 100%.

Table 10: Statistics about the deciles of parameter changed ranges of FT and PT LLMs.

| Models | 0% (min) | 10% | 20% | 30% | 40% | 50% | 60% | 70% | 80% | 90% | 100% (max) |
|---|---|---|---|---|---|---|---|---|---|---|---|
| Qwen1.5-1.8B-Chat vs. Qwen1.5-1.8B | -0.10 | -0.29e-02 | -0.19e-02 | -0.11e-02 | -0.05e-02 | 0.00 | 0.05e-02 | 0.11e-02 | 0.19e-02 | 0.29e-02 | 0.14 |
| Sailor-1.8B vs. Qwen1.5-1.8B | -6.25e-02 | -1.00e-02 | -0.51e-02 | -0.23e-02 | -0.06e-02 | 0.00 | 0.06e-02 | 0.23e-02 | 0.51e-02 | 1.00e-02 | 6.25e-02 |
| Qwen1.5-4B-Chat vs. Qwen1.5-4B | -2.34e-02 | -4.88e-04 | -2.75e-04 | -1.83e-04 | -7.63e-05 | 0.00 | 7.63e-05 | 1.83e-04 | 2.75e-04 | 4.88e-04 | 1.90e-02 |
| Sailor-4B vs. Qwen1.5-4B | -0.63 | -0.96e-02 | -0.62e-02 | -0.38e-02 | -0.18e-02 | 0.00 | 0.18e-02 | 0.38e-02 | 0.62e-02 | 0.96e-02 | 0.63 |
| Qwen1.5-7B-Chat vs. Qwen1.5-7B | -2.43e-02 | -4.27e-04 | -2.44e-04 | -1.22e-04 | -3.05e-05 | 0.00 | 3.05e-05 | 1.22e-04 | 2.44e-04 | 4.27e-04 | 2.29e-02 |
| Sailor-7B vs. Qwen1.5-7B | -0.27 | -0.57e-02 | -0.37e-02 | -0.23e-02 | -0.11e-02 | 0.00 | 0.11e-02 | 0.23e-02 | 0.37e-02 | 0.57e-02 | 0.25 |
| Qwen1.5-14B-Chat vs. Qwen1.5-14B | -2.34e-02 | -4.27e-04 | -2.44e-04 | -1.22e-04 | -3.05e-05 | 0.00 | 3.05e-05 | 1.22e-04 | 2.44e-04 | 4.27e-04 | 2.06e-02 |
| Sailor-14B vs. Qwen1.5-14B | -0.36 | -0.78e-02 | -0.51e-02 | -0.31e-02 | -0.15e-02 | 0.00 | 0.15e-02 | 0.31e-02 | 0.51e-02 | 0.78e-02 | 0.42 |
| WizardLM-13B vs. Llama-2-13b | -3.93e-02 | -0.16e-02 | -0.10e-02 | -0.06e-02 | -0.03e-02 | 0.00 | 0.03e-02 | 0.06e-02 | 0.10e-02 | 0.16e-02 | 4.81e-02 |
| WizardMath-13B vs. Llama-2-13b | -0.69e-02 | -0.06e-02 | -0.04e-02 | -0.02e-02 | -0.01e-02 | 0.00 | 0.01e-02 | 0.02e-02 | 0.04e-02 | 0.06e-02 | 0.74e-02 |
| llama-2-13b-code-alpaca vs. Llama-2-13b | -8.42e-02 | -3.05e-05 | 0.00 | 0.00 | 0.00 | 0.00 | 0.00 | 0.00 | 0.00 | 3.05e-05 | 7.98e-02 |
| finance-chat vs. Llama-2-7b-chat | -3.78e-02 | -3.66e-04 | -3.05e-05 | 0.00 | 0.00 | 0.00 | 0.00 | 0.00 | 3.05e-05 | 3.66e-04 | 5.07e-02 |
| medicine-chat vs. Llama-2-7b-chat | -3.79e-02 | -0.03e-02 | 0.00 | 0.00 | 0.00 | 0.00 | 0.00 | 0.00 | 0.00 | 0.03e-02 | 5.03e-02 |
| law-chat vs. Llama-2-7b-chat | -3.61e-02 | -0.03e-02 | 0.00 | 0.00 | 0.00 | 0.00 | 0.00 | 0.00 | 0.00 | 0.03e-02 | 4.77e-02 |
| BioMistral-7B vs. Mistral-7B-Instruct-v0.1 | -6.25e-02 | -0.11e-02 | -0.07e-02 | -0.04e-02 | -0.02e-02 | 0.00 | 0.02e-02 | 0.04e-02 | 0.07e-02 | 0.11e-02 | 1.86e-02 |
| Saul-7B-Base vs. Mistral-7B-v0.1 | -4.40e-03 | -1.22e-04 | -7.63e-05 | -4.58e-05 | -2.48e-05 | 0.00 | 2.48e-05 | 4.58e-05 | 7.63e-05 | 1.22e-04 | 4.15e-03 |

## A.8 ADDITIONAL RESULTS OF MERGING QWEN1.5-CHAT AND SAILOR ACROSS 1.8B AND 4B MODEL SIZES

Table 11 and Table 12 show the performance of merging Qwen1.5-Chat and Sailor on South-East Asian language benchmark and Open LLM Leaderboard across 1.8B and 4B model sizes.

Table 11: Performance of merging Qwen1.5-Chat and Sailor on South-East Asian language benchmark across 1.8B and 4B model sizes.

| Size | Models | Merging Methods | XQuAD th | TydiQA id | XQuAD vi | XCOPA th | XCOPA id | XCOPA vi | Belebele th | Belebele id | Belebele vi | M3Exam jv | Average | Average Rank |
|---|---|---|---|---|---|---|---|---|---|---|---|---|---|---|
| | Qwen1.5 | / | 27.24/43.56 | 29.73/53.76 | 29.17/48.15 | 52.60 | 51.60 | 53.40 | 30.11 | 32.00 | 31.33 | 24.26 | 38.99 | / |
| | Qwen1.5-Chat | / | 18.10/31.43 | 24.42/49.10 | 24.64/43.13 | 53.00 | 53.20 | 54.40 | 29.89 | 32.00 | 34.00 | 26.15 | 36.42 | / |
| | Sailor | / | 32.72/48.66 | 40.88/65.37 | 34.22/53.35 | 53.80 | 64.20 | 63.20 | 34.22 | 34.89 | 35.33 | 28.30 | 45.32 | / |
| 1.8B | | Task Arithmetic | 36.81/51.43 | 33.81/62.82 | 32.68/52.62 | 55.00 | **65.40** | 59.80 | **34.33** | 36.22 | **36.11** | **28.30** | 45.03 | 1.85 |
| | | SLERP | 28.37/44.64 | 21.77/53.76 | 29.26/51.39 | 54.40 | 54.40 | 57.40 | 32.22 | 34.33 | 35.44 | 27.22 | 40.35 | 4.15 |
| | Qwen1.5-Chat | Model Stock | 28.63/44.35 | 30.97/56.50 | 31.65/51.14 | 52.80 | 51.60 | 54.80 | 30.89 | 33.00 | 31.44 | 23.99 | 40.14 | 4.85 |
| | & Sailor | Breadcrumbs | 22.45/31.95 | 20.18/43.83 | 25.49/42.11 | 53.40 | 57.40 | 59.80 | 31.56 | 34.67 | 34.89 | 27.22 | 37.30 | 4.92 |
| | | TIES-Merging | 26.02/41.09 | 36.81/61.68 | 31.99/52.40 | 52.00 | 62.60 | **60.40** | 33.78 | **36.89** | 35.89 | 25.61 | 42.86 | 3.15 |
| | | WIDEN | **38.21/53.50** | **43.36/68.55** | **37.55/56.05** | **55.20** | 61.80 | 60.20 | 34.22 | 35.33 | 36.00 | 27.49 | **46.73** | **1.62** |
| | Qwen1.5 | / | 34.03/53.40 | 48.32/72.68 | 43.71/63.86 | 53.40 | 55.00 | 57.80 | 32.78 | 36.22 | 35.22 | 24.26 | 46.98 | / |
| | Qwen1.5-Chat | / | 27.76/41.84 | 44.96/66.09 | 39.95/59.46 | 51.20 | 52.80 | 53.60 | 34.11 | 39.33 | 37.44 | 24.80 | 44.10 | / |
| | Sailor | / | 46.82/63.34 | 53.98/73.48 | 47.65/67.09 | 53.40 | 69.20 | 68.20 | 36.11 | 41.33 | 38.89 | 31.27 | 53.14 | / |
| 4B | | Task Arithmetic | **28.98/45.21** | 16.28/28.27 | 19.76/36.27 | 53.80 | 60.40 | 58.40 | 34.11 | 39.11 | 36.89 | 23.99 | 37.04 | 2.85 |
| | | SLERP | 11.92/28.09 | 19.47/42.16 | **31.74/52.56** | 51.40 | 57.00 | 56.60 | 33.33 | 39.44 | **38.22** | 25.88 | 37.52 | 2.54 |
| | Qwen1.5-Chat | Model Stock | 10.27/26.73 | 16.64/47.73 | 30.37/**52.69** | 51.00 | 53.00 | 58.00 | 31.89 | 38.56 | 37.11 | **27.22** | 37.02 | 3.08 |
| | & Sailor | Breadcrumbs | 0.70/1.80 | 5.49/9.14 | 1.54/1.67 | 48.80 | 56.20 | 55.80 | 28.33 | 29.11 | 30.56 | 24.80 | 22.61 | 4.92 |
| | | TIES-Merging | 0.00/0.50 | 0.18/2.86 | 0.43/1.13 | 52.00 | 53.00 | 52.80 | 26.44 | 29.56 | 29.11 | 24.53 | 20.96 | 5.46 |
| | | WIDEN | 25.67/45.08 | **20.00/48.80** | 25.49/42.17 | **54.00** | **63.40** | **58.80** | **35.89** | **42.00** | 33.22 | 24.53 | **39.93** | **1.92** |

Table 12: Performance of merging Qwen1.5-Chat and Sailor on Open LLM Leaderboard across 1.8B and 4B model sizes.

| Size | Models | Merging Methods | ARC | Hella-Swag | MMLU | Truthful-QA | Wino-grande | GSM8K | Average | Average Rank |
|---|---|---|---|---|---|---|---|---|---|---|
| 1.8B | Qwen1.5 | / | 37.80 | 61.67 | 45.71 | 39.33 | 61.64 | 34.04 | 46.70 | / |
| | Qwen1.5-Chat | / | 39.68 | 60.36 | 44.53 | 40.57 | 59.83 | 31.39 | 46.06 | / |
| | Sailor | / | 32.59 | 57.48 | 29.60 | 37.77 | 59.98 | 2.65 | 36.68 | / |
| | Qwen1.5-Chat & Sailor | Task Arithmetic | 37.20 | 60.43 | 41.45 | 38.95 | 61.96 | 12.74 | 42.12 | 4.83 |
| | | SLERP | **39.51** | 61.17 | 43.96 | **40.95** | 60.85 | 25.40 | 45.31 | 2.17 |
| | | Model Stock | 37.97 | **61.82** | **46.23** | 39.84 | 61.96 | **34.50** | **47.05** | **1.67** |
| | | Breadcrumbs | 37.80 | 60.56 | 41.44 | 38.36 | **62.04** | 17.36 | 42.93 | 3.50 |
| | | TIES-Merging | 37.54 | 60.56 | 41.13 | 39.39 | 61.72 | 14.25 | 42.41 | 4.50 |
| | | WIDEN | 37.71 | 60.47 | 41.61 | 40.54 | 61.64 | 13.04 | 42.50 | 3.67 |
| 4B | Qwen1.5 | / | 48.04 | 71.43 | 55.01 | 47.22 | 68.43 | 52.31 | 57.07 | / |
| | Qwen1.5-Chat | / | 43.26 | 69.67 | 54.07 | 44.74 | 66.61 | 5.84 | 47.37 | / |
| | Sailor | / | 44.45 | 69.38 | 36.80 | 37.03 | 65.35 | 11.75 | 44.13 | / |
| | Qwen1.5-Chat & Sailor | Task Arithmetic | 46.50 | 64.01 | 38.25 | 43.73 | 65.19 | 8.49 | 44.36 | 4.00 |
| | | SLERP | 45.56 | 68.25 | 50.01 | 43.88 | 66.38 | 41.70 | 52.63 | 2.83 |
| | | Model Stock | **47.01** | **69.31** | **55.41** | 46.55 | **67.32** | **47.08** | **55.45** | **1.33** |
| | | Breadcrumbs | 39.16 | 43.15 | 43.84 | 48.55 | 61.80 | 0.00 | 39.42 | 4.33 |
| | | TIES-Merging | 35.15 | 41.04 | 30.15 | **49.47** | 59.19 | 0.00 | 35.83 | 5.00 |
| | | WIDEN | 45.90 | 66.05 | 48.66 | 43.34 | 66.69 | 13.95 | 47.43 | 3.33 |

A.9  ETHICS STATEMENT

This work investigates the merging task of LLMs, no matter they are fine-tuned or pre-trained models. Even though this work has no direct ethical problems, LLMs may still potentially generate harmful information including gender bias, fake news, and private messages when equipped with our approach. It is necessary and promising to design specialized mechanisms to carefully regulate these underlying issues.

A.10  REPRODUCIBILITY STATEMENT

We ensure the reproducibility of this work by presenting the experimental details in Section 4.1 and Appendix. Additionally, implementation of the proposed algorithm is available at `https://anonymous.4open.science/r/MergeLLM-5E0D`.

