# OpenReview forum: "Extend Model Merging from Fine-Tuned to Pre-Trained Large Language Models via Weight Disentanglement"
_ICLR.cc/2025/Conference — Submitted to ICLR 2025_

### Official Review · Reviewer_3uPt · 2024-10-31

**Soundness:** 3
**Presentation:** 2
**Contribution:** 2
**Rating:** 5
**Confidence:** 3

**Summary:**

The paper presents WIDEN, a novel merging technique for Large Language Models (LLMs), which extends the applicability of merging from finetuned (FT) to pretrained (PT) models by disentangling weights into magnitude and direction components. This weight disentanglement enables adaptive merging by quantifying each LLM's alteration relative to a shared backbone. The disentangled weights are ranked using normalized divergence scores compared to the pretrained baseline, and this ranking is used to compute an automated importance factor for each LLM. This results in a generalized form of several existing arithmetic methods for LLM merging. Experimental results suggest that WIDEN effectively balances multiple capabilities, such as multilingual and instruction-following skills, across FT and PT LLMs.

**Strengths:**

- The paper’s effort to expand merging capabilities from FT to PT models is well-motivated and addresses a crucial gap in existing merging techniques.
- The methodology has a sound technical foundation, with a detailed four-step framework integrating weight disentanglement, ranking, and adaptive score calibration.
- The experimental setup is thorough, covering both conventional FT merging tasks and the new FT-PT merging setting. WIDEN’s performance across SEA and Open LLM Leaderboard benchmarks and comparison with multiple baselines highlights its applicability to diverse LLMs.
- The impact of each component within WIDEN is evaluated with an ablation experiment in Figure 2, demonstrating the importance of weight disentanglement and score calibration.

**Weaknesses:**

- Although WIDEN generalizes across FT and PT models, it does not consistently outperform Task Arithmetic on all benchmarks. For instance, Task Arithmetic often shows competitive results on Open LLM Leaderboard tasks, raising concerns about WIDEN’s scalability and stability. For example, on the SEA benchmark, the performance improvement on 14B models is smaller than the 7B model, with the gap between Task Arithmetic and its claimed generalized form WIDEN narrowing as the LLMs become larger.
- The improvement WIDEN demonstrates is noticeably higher on SEA benchmarks than on the OpenLLM Leaderboard, yet the paper does not clarify why performance fluctuates between benchmarks. This omission raises questions about its adaptability to different domains or task settings.
- While grid search is used for tuning, the choice of hyperparameters (particularly t and s) lacks justification beyond empirical results. A clearer rationale or theoretical insight into their selection would enhance the robustness of WIDEN’s methodology.
- Although score calibration is a novel addition to ensure adaptive ranking and merging, values other than 1.0 should be evaluated in score calibration. The "ease of implementation" rationale is not good enough.

**Questions:**

- How does WIDEN handle cases where the backbone (reference pretrained model) diverges substantially in structure or task specificity from both FT and PT models? Would WIDEN work with heterogeneous LLMs beyond those sharing the same backbone?
- Did the authors attempt to merge more than two or three models to evaluate WIDEN’s scalability and robustness? If so, what were the results, and how does performance change as the number of LLMs increases?
- Given WIDEN’s better performance on SEA benchmarks than on the OpenLLM Leaderboard, could the authors elaborate on why this discrepancy exists? Is WIDEN more suited to particular types of tasks or linguistic benchmarks?
- On tasks where Task Arithmetic performs better, why might WIDEN’s performance lag?
- Since WIDEN modifies weights adaptively, would it be feasible to incorporate it into a continual learning setup where multiple LLMs are progressively merged over time? Could this method be used for models other than LLMs?

---

> ### Author Response · Authors · 2024-11-22
> **Response to Reviewer 3uPt (Part 1/4)**
>
> Thanks for your constructive comments. We have undertaken several revisions to address your concerns. Firstly, we have analyzed the varying performance gap between WIDEN and baselines across different benchmarks. Secondly, we have explained the rationale behind setting hyperparameters $t$ and $s$ and provided empirical validation of WIDEN’s performance under different settings of $t$ and $s$. Thirdly, we have compared the performance of WIDEN and baselines when merging three FT LLMs. Finally, we have discussed the applicability of WIDEN in merging heterogeneous LLMs, continual learning setups, and models beyond LLMs. We hope our answers have adequately addressed your concerns. We are happy to explain more if further discussions are required.
>
> **W1: Although WIDEN generalizes across FT and PT models, it does not consistently outperform Task Arithmetic on all benchmarks. For instance, Task Arithmetic often shows competitive results on Open LLM Leaderboard tasks, raising concerns about WIDEN’s scalability and stability. On the SEA benchmark, the performance improvement on 14B models is smaller than the 7B model, with the gap between Task Arithmetic and its claimed generalized form WIDEN narrowing as the LLMs become larger.**
>
> Thank you for raising this point. When evaluating model performance, it is recommended to consider both average performance and average rank metrics. On the South-East Asian language benchmark, although the average rank for the 14B model is 1.77 (compared to 1.15 for the 7B model), the average performance gap between WIDEN and the best baseline on the 14B model is 10.15 (59.67 vs. 49.52), which is more significant than the 5.20 gap observed for the 7B model (56.27 vs. 51.07). Regarding the Open LLM Leaderboard, we agree that WIDEN achieves competitive but not state-of-the-art performance compared with baselines. However, when considering both benchmarks together, WIDEN consistently delivers satisfactory results across both, while most baselines fail to do so, demonstrating the robustness and generalizability of WIDEN. These explanations have been added to Section 4.2 in the revised version.
>
> Additionally, we have incorporated the merging results of Sailor and Qwen1.5-Chat at 1.8B and 4B scales for more comprehensive comparisons. The results show that WIDEN outperforms baselines on the South-East Asian language benchmark, and its performance on the Open LLM Leaderboard improves as the size of the LLMs increases. Please refer to our response to **Reviewer W7XB’s W4** for more details. We have added these contents in Section 4.2 and Appendix Section A.8 in the revised manuscript.
>
> Lastly, we want to clarify that for baselines like Task Arithmetic that rely on scaling terms, we select the optimal setting at the dataset level within the range [0.5, 1.0], rather than using an identical setting at the model level. For instance, see the following results for 7B models. On the Open LLM Leaderboard, three datasets (ARC, MMLU, and Winogrande) perform better with a scaling term of 0.5, while the other three datasets (HellaSwag, TruthfulQA, and GSM8K) perform better with 1.0. While on the South-East Asian language benchmark, a scaling term of 1.0 consistently outperforms 0.5. For WIDEN, we aim to compute the importance of weights through weight disentanglement, eliminating the need for manual specification. Even for hyperparameters $t$ and $s$, we used a unified setting across all benchmarks. Such an implementation may reduce the advantage of WIDEN on the Open LLM Leaderboard to some extent but demonstrates its robustness and generalizability. Our response to your **W4** shows that searching fine-grained settings of $t$ and $s$ can yield better results. We have added the above contents in Appendix Section A.5 in the revised version.
>
> | **Open LLM Leaderboard** | **ARC** | **HellaSwag** | **MMLU** | **TruthfulQA** | **Winogrande** | **GSM8K** |
> | --- | --- | --- | --- | --- | --- | --- |
> | scaling term = 0.5 | **52.05** | 68.42 | **59.38** | 46.94 | **69.77** | 19.41 |
> | scaling term = 1.0 | 50.51 | **75.15** | 51.47 | **50.84** | 68.19 | **25.55** |
>
> | **South-East Asian language benchmark** | **XQuAD (th)** | **TydiQA (id)** | **XQuAD (vi)** | **XCOPA (th)** | **XCOPA (id)** | **XCOPA (vi)** | **Belebele (th)** | **Belebele (id)** | **Belebele (vi)** | **M3Exam (jv)** |
> | --- | --- | --- | --- | --- | --- | --- | --- | --- | --- | --- |
> | scaling term = 0.5 | 1.74/12.76 | 2.83/13.63 | 10.27/22.87 | 54.40 | 70.40 | 66.80 | 35.56 | 44.56 | 42.11 | 28.03 |
> | scaling term = 1.0 | **28.20**/**49.62** | **45.84**/**65.78** | **37.38**/**61.53** | **63.20** | **77.60** | **73.40** | **38.89** | **46.89** | **45.11** | **30.46** |

---

> > ### Author Response · Authors · 2024-11-22
> > **Response to Reviewer 3uPt (Part 2/4)**
> >
> > **W2: The improvement WIDEN demonstrates is noticeably higher on SEA benchmark than on the OpenLLM Leaderboard, yet the paper does not clarify why performance fluctuates between benchmarks. This omission raises questions about its adaptability to different domains or task settings.**
> >
> > As addressed in our response to **W1**, to validate the robustness and generalizability of WIDEN, we select hyperparameters at the model level rather than the dataset level. This implementation may reduce the performance gap between WIDEN and baselines in some cases but WIDEN still achieves competitive or better results under such a setup. Since the baselines cannot benefit from integrating the performance of scaling terms 0.5 and 1.0 on the South-East Asian language benchmark, WIDEN depicts a significant advantage on this benchmark compared to the Open LLM Leaderboard.
> >
> > **W3: The choice of hyperparameters (particularly t and s) lacks justification beyond empirical results. A clearer rationale or theoretical insight into their selection would enhance the robustness of WIDEN’s methodology.**
> >
> > As mentioned in Remark 2 in Section 3.3, Average Merging and Task Arithmetic can be viewed as special cases of WIDEN by setting $t < 0.0$ and  $s= 1 / N$, and  $t < 0.0$ and $s= \lambda$, respectively. Note that $t$ and $s$ are designed to control the merging importance of parameters after applying the Softmax function. Specifically, if we want to assign higher importance to more parameters, $t$ should be reduced and $s$ should be increased. Conversely, $t$ should be increased and $s$ should be reduced. This provides a rationale for choosing $t$ and $s$ based on the desired parameter requirements. We have added this explanation in Section 3.3 in the revised manuscript.
> >
> > **W4: Although score calibration is a novel addition to ensure adaptive ranking and merging, values other than 1.0 should be evaluated in score calibration. The "ease of implementation" rationale is not good enough.**
> >
> > We have explained how to select the values of $t$ and $s$ based on parameter requirements in response to **W3**. Here, we further empirically show the model performance under different settings of $t$ and $s$ for the 7B model size. Results indicate that more refined settings  for $t$ and $s$ in score calibration can lead to performance improvements. For example, on the Open LLM Leaderboard, when $s=1.0$, setting $t=1.5$ improves performance on ARC, MMLU, Winogrande, and GSM8K, and $t=0.0$ outperforms the default $t=1.0$ setting on TruthfulQA. Additionally, different values of $s$ when $t=1.0$ also show improvement over the default setting of $s=1.0$ on certain benchmarks. On the South-East Asian language benchmark, when $s=1.0$, the settings of $t=0.5$ or $t=1.5$, as well as $s=0.9$ and $s=1.1$ when $t=1.0$, show superior performance over the setting of $t=1.0$ and $s=1.0$ on datasets like XQuAD (th), TydiQA (id), XQuAD (vi), XCOPA (id), XCOPA (vi), and Belebele (th). We ultimately set $s=1.0$ across all the datasets for ease of implementation. However, It is feasible to explore more optimal settings of $t$ or $s$ if higher metrics are desired.

---

> > > ### Author Response · Authors · 2024-11-22
> > > **Response to Reviewer 3uPt (Part 3/4)**
> > >
> > > | **Open LLM Leaderboard** | **ARC** | **HellaSwag** | **MMLU** | **TruthfulQA** | **Winogrande** | **GSM8K** |
> > > | --- | --- | --- | --- | --- | --- | --- |
> > > | $t=0.0$, $s=1.0$ | 50.43 | 74.94 | 51.46 | **50.66** | 68.19 | 26.99 |
> > > | $t=0.5$, $s=1.0$ | 52.47 | 75.21 | 54.98 | 49.85 | 70.56 | 38.13 |
> > > | $t=1.0$, $s=1.0$ (default) | 53.84 | **76.25** | 57.65 | 49.34 | 71.90 | 44.81 |
> > > | $t=1.5$, $s=1.0$ | **54.95** | 75.16 | **58.58** | 49.29 | **73.09** | **46.70** |
> > > | $t=2.0$, $s=1.0$ | 52.47 | 68.36 | 58.46 | 46.70 | 70.56 | 18.42 |
> > > |  |  |  |  |  |  |  |
> > > | $t=1.0$, $s=0.5$ | 51.96 | 68.35 | **59.47** | 46.53 | 70.09 | 22.59 |
> > > | $t=1.0$, $s=0.8$ | **54.27** | 73.50 | 58.64 | 49.02 | 71.90 | 45.41 |
> > > | $t=1.0$, $s=0.9$ | 53.83 | 75.19 | 58.09 | 49.30 | 71.98 | **46.47** |
> > > | $t=1.0$, $s=1.0$ (default) | 53.84 | **76.25** | 57.65 | 49.34 | 71.90 | 44.81 |
> > > | $t=1.0$, $s=1.1$ | 53.58 | 76.14 | 56.67 | 49.42 | **72.14** | 44.50 |
> > > | $t=1.0$, $s=1.2$ | 52.56 | 75.67 | 55.40 | 49.49 | 71.43 | 39.73 |
> > > | $t=1.0$, $s=1.5$ | 45.82 | 65.15 | 48.28 | **50.18** | 66.38 | 5.31 |
> > >
> > > | **South-East Asian language benchmark** | **XQuAD (th)** | **TydiQA (id)** | **XQuAD (vi)** | **XCOPA (th)** | **XCOPA (id)** | **XCOPA (vi)** | **Belebele (th)**  | **Belebele (id)** | **Belebele (vi)** | **M3Exam (jv)** |
> > > | --- | --- | --- | --- | --- | --- | --- | --- | --- | --- | --- |
> > > | $t=0.0$, $s=1.0$ | 27.59/49.80          | 45.49/65.94 | 38.24/62.97 | 62.60 | 77.60 | 73.60 | 38.22 | 46.11 | 45.44 | 31.27 |
> > > | $t=0.5$, $s=1.0$ | **47.61**/**67.27**          | **52.92**/**75.70** | **49.96**/**73.65** | 59.00 | **77.60** | 73.80 | **41.00** | **51.22** | 48.00 | 31.27 |
> > > | $t=1.0$, $s=1.0$ (default) | 42.65/64.21 | 45.84/73.37 | 48.42/73.17 | **60.20** | 77.40 | 73.60 | 40.11 | 51.11 | **48.56** | **32.88** |
> > > | $t=1.5$, $s=1.0$ | 21.58/48.03 | 21.24/57.00 | 39.18/67.79 | 58.80 | 75.60 | **74.20** | 39.22 | 50.00 | 47.11 | 31.00 |
> > > | $t=2.0$, $s=1.0$ | 4.44/16.78 | 4.60/15.36 | 12.23/24.78 | 54.20 | 71.40 | 67.20 | 35.67 | 44.44 | 42.67 | 29.38 |
> > > |  |  |  |  |  |  |  |  |  |  |  |
> > > | $t=1.0$, $s=0.5$ | 2.87/14.26 | 4.42/15.67 | 12.06/24.80 | 54.40 | 70.60 | 66.60 | 35.11 | 43.89 | 41.56 | 27.76 |
> > > | $t=1.0$, $s=0.8$ | 13.66/40.34 | 15.04/47.51 | 33.79/60.81 | 57.00    | 74.20 | 73.40 | 37.78 | 49.56 | 45.89 | 31.27 |
> > > | $t=1.0$, $s=0.9$ | 23.50/50.84          | 27.26/62.39 | 44.05/71.09 | 58.80 | 76.00 | **73.80** | **40.44** | 50.56 | 46.78 | 31.81 |
> > > | $t=1.0$, $s=1.0$ (default) | 42.65/**64.21** | 45.84/73.37 | 48.42/**73.17** | **60.20** | 77.40 | 73.60 | 40.11 | **51.11** | **48.56** | **32.88** |
> > > | $t=1.0$, $s=1.1$ | **44.56**/64.04         | **58.58**/**77.28**  | **49.79**/72.68 | 59.20 | **78.00** | 73.00 | 40.11 | 49.78 | 46.89 | 30.46 |
> > > | $t=1.0$, $s=1.2$ | 31.07/52.99          | 51.50/70.82 | 41.83/63.61 | 58.60 | 76.40 | 72.60 | 37.78 | 47.56 | 42.89 | 30.73 |
> > > | $t=1.0$, $s=1.5$ | 2.79/3.69         | 3.01/7.26  | 5.65/11.71 | 58.80 | 71.00 | 63.80 | 31.56 | 35.33 | 31.44 | 29.65 |

---

> > > > ### Author Response · Authors · 2024-11-22
> > > > **Response to Reviewer 3uPt (Part 4/4)**
> > > >
> > > > **Q1: How does WIDEN handle cases where the backbone (reference pre-trained model) diverges substantially in structure or task specificity from both FT and PT models? Would WIDEN work with heterogeneous LLMs beyond those sharing the same backbone?**
> > > >
> > > > If the backbone and the FT or PT models substantially differ from the architecture, they can be considered as heterogeneous models. Currently, WIDEN is only designed for merging homologous LLMs that share the same backbone, which is a standard setting for existing merging methods [1-4]. Merging models with different architectures is challenging due to parameter misalignment. It is worth noting that there are emerging studies that explore merging heterogeneous models from a probabilistic distribution perspective [5, 6]. We believe that extending WIDEN to handle heterogeneous LLMs is an intriguing direction for future research. For example, enabling the fusion of target LLMs in approaches like FUSELLM [5] or FUSECHAT [6] through WIDEN.
> > > >
> > > > From a task-specific perspective, most existing FT and PT models have not undergone sufficient training that causes significant divergence in parameters compared to their backbones (see our response to **Reviewer W7XB’s W1**). Therefore, we have not encountered cases where the parameter changes of both FT and PT LLMs are substantially different from the backbone.
> > > >
> > > > [1] Matena M S, Raffel C A. Merging models with fisher-weighted averaging. 2022, NeurIPS.
> > > >
> > > > [2] Ilharco G, Ribeiro M T, Wortsman M, et al. Editing models with task arithmetic. 2023, ICLR.
> > > >
> > > > [3] Yadav P, Tam D, Choshen L, et al. Ties-merging: Resolving interference when merging models. 2023, NeurIPS.
> > > >
> > > > [4] Yu L, Yu B, Yu H, et al. Language models are super mario: Absorbing abilities from homologous models as a free lunch. ICML, 2024.
> > > >
> > > > [5] Wan F, Huang X, Cai D, et al. Knowledge fusion of large language models. 2024, ICLR.
> > > >
> > > > [6] Wan F, Zhong L, Yang Z, et al. Fusechat: Knowledge fusion of chat models. 2024, arXiv.
> > > >
> > > > **Q2: Did the authors attempt to merge more than two or three models to evaluate WIDEN’s scalability and robustness? If so, what were the results, and how does performance change as the number of LLMs increases?**
> > > >
> > > > We have attempted to merge three LLMs (WizardLM-13B, WizardMath-13B, and llama-2-13b-code-alpaca). The results indicate that WIDEN can achieve a good balance of three abilities, particularly in code generation, which is consistent with the observation when merging two FT models. For further details, please refer to our response to **Reviewer KKeW’s W2**. We have incorporated the above findings in Section 4.3 (Table 5) in the revised version. Due to the limited time during rebuttal, we are unable to explore the impact of the number of LLMs to be merged on performance, which may serve as a direction for future work.
> > > >
> > > > **Q3 & Q4: Given WIDEN’s better performance on SEA benchmark than on the OpenLLM Leaderboard, could the authors elaborate on why this discrepancy exists? Is WIDEN more suited to particular types of tasks or linguistic benchmarks? On tasks where Task Arithmetic performs better, why might WIDEN’s performance lag?**
> > > >
> > > > We appreciate these two constructive comments. As detailed in our response to **W1** and **W2**, the performance discrepancy mainly arises due to the implementation of the grid search. Please see our response to **W1** and **W2** for more detailed information.
> > > >
> > > > **Q5: Since WIDEN modifies weights adaptively, would it be feasible to incorporate it into a continual learning setup where multiple LLMs are progressively merged over time? Could this method be used for models other than LLMs?**
> > > >
> > > > Yes. Technically, WIDEN can be applied in a continual learning setup where models are merged progressively over time, which is similar to Robust Fine-Tuning [1]. Moreover, the applicability of WIDEN is not inherently tied to LLMs and can be applied to other model types, as long as they share the same backbone. We believe that exploring the use of WIDEN in continual learning scenarios or with various model architectures (such as general language models, vision models, and other types of neural networks) is a promising direction for future research.
> > > >
> > > > [1] Wortsman M, Ilharco G, Kim J W, et al. Robust fine-tuning of zero-shot models. 2022, CVPR.

---

> > > > > ### Author Response · Authors · 2024-11-27
> > > > > **Gentle Reminder to Review Rebuttal Response of Submission 1006**
> > > > >
> > > > > Dear Reviewer 3uPt,
> > > > >
> > > > > Thanks for your time and valuable feedback on our submission. We have received positive scores (6 and 6) from two reviewers. We are writing to kindly remind you that we have responded to your questions and would appreciate it if you could review our response when you have the opportunity. We believe that we have addressed the concerns you raised. Please let us know if you have any remaining issues or questions.
> > > > >
> > > > > We appreciate your efforts in reviewing our work and look forward to your feedback.
> > > > >
> > > > > Best regards,
> > > > >
> > > > > Authors of submission 1006

---

> ### Comment · Area_Chair_vaBV · 2024-11-30
>
> Dear Reviewer,
>
> Thank you for your efforts in reviewing the submission. The authors have provided some feedback. We kindly encourage you to review the responses to see if they can address your concerns. Your timely input is critical for the next steps in the review process.
>
> Best regards,
>
> AC

---

### Official Review · Reviewer_KKeW · 2024-11-02

**Soundness:** 3
**Presentation:** 4
**Contribution:** 3
**Rating:** 6
**Confidence:** 2

**Summary:**

This paper presents a pioneering effort in extending model merging to Pretrained LLMs utilizing weight disentanglement. Extensive studies on previous methods demonstrate their inability to perform when applied to pretrained LLM while the method proposed in the paper is able to solve the task with minimal performance drop compared to the models to be merged.

**Strengths:**

1. The paper is the first successful attempt to incorporate the ability of PT LLM into model merging techniques.
2. Extensive experiments and analyses have demonstrated the effectiveness of the proposed method.
3. The paper is well-written and easy to follow.

**Weaknesses:**

1. The experiments are only limited to Sailor, more results on different models could validate the effectiveness of the proposed method.
2. Despite being indicated by the method, the experiments didn't show evidence that the proposed method could work for multiple LLM cases. Some experiments from this perspective would be appreciated.

**Questions:**

See weaknesses.

I'm not familiar with LLM merging and am open to discussion if misunderstood any part of the paper.

---

> ### Author Response · Authors · 2024-11-22
> **Response to Reviewer KKeW**
>
> Thanks for the valuable comments. We have addressed each of your concerns as follows. Firstly, we have expanded the merging experiments to include Sailor and Qwen1.5-Chat with 1.8B and 4B parameters and pointed out the common issue of insufficient pre-training in most existing PT LLMs. Secondly, we have provided the comparisons of merging three FT LLMs to evaluate the performance of WIDEN in such scenarios. More discussions are welcomed if there are any further problems.
>
> **W1: The experiments are only limited to Sailor, more results on different models could validate the effectiveness of the proposed method.**
>
> We understand your concern regarding the limited scope of our initial experiments. To address this, we have extended the merging experiments of Sailor and Qwen1.5-Chat to 1.8b and 4B sizes to make the comparisons more comprehensive. The results demonstrate that WIDEN consistently performs well across different model sizes, effectively integrating the multilingual capabilities of Sailor with the general abilities of Qwen1.5-Chat. Please see our response to **Reviewer W7XB’s W4** for more details. We have added these contents in Section 4.2 and Appendix Section A.8 in the revised manuscript.
>
> Additionally, we have investigated several existing PT LLMs, including finance-chat, medicine-chat, law-chat, BioMistral-7B, and Saul-7B-Base. However, these models are pre-trained on fewer than 30B tokens, resulting in relatively small parameter changed ranges. This makes them less suitable for our experimental setup, as substantial parameter changes are desired for the experiments. These discussion is discussed in our response to **Reviewer W7XB’s W1** and has been included in Appendix Sections A.6 and A.7 of the revised manuscript.
>
> Currently, we believe it is challenging to acquire LLMs with significantly different capabilities that have undergone both fine-tuning and sufficient pre-training on the same backbone. Thus, it is challenging for us to add additional merging experiments with other models during the rebuttal period. We are happy to conduct more experiments if you could kindly provide information on LLMs that meet the criteria.
>
> **W2: Despite being indicated by the method, the experiments didn't show evidence that the proposed method could work for multiple LLM cases. Some experiments from this perspective would be appreciated.**
>
> Thanks for this valuable suggestion. We have merged three FT LLMs (WizardLM-13B, WizardMath-13B, and llama-2-13b-code-alpaca) and show the results as follows.
>
> | **Merging Methods** | **AlpacaEval 2.0** | **GSM8K** | **MATH** | **HumanEval** | **MBPP** |
> | --- | --- | --- | --- | --- | --- |
> | Task Arithmetic | **11.51** | 58.45 | 9.88 | 18.29 | 29.80 |
> | Model Stock | 0.12 | 0.00 | 0.00 | 5.49 | 23.40 |
> | TIES-Merging | 9.22 | **62.55** | 9.54 | 21.95 | 30.40 |
> | Breadcrumbs | 10.89 | **62.55** | **10.58** | **23.78** | 29.60 |
> | WIDEN | 8.71 | 57.16 | 9.60 | 22.56 | **30.80** |
>
> We find that WIDEN still achieves a balanced amalgamation of three abilities (especially in code generation). Though the advantage of WIDEN is less pronounced on merging multiple FT LLMs, we would like to emphasize that the strength of WIDEN lies in merging LLMs with substantial differences in parameter changed ranges, which significantly outperforms previous methods in merging PT and FT LLMs. We have added the above results in Section 4.3 (Table 5) in the revised manuscript.

---

> > ### Comment · Reviewer_KKeW · 2024-11-25
> >
> > Thank the authors for the point-by-point response. The responses have addressed my concerns. I decide to maintain my positive score.

---

> > > ### Author Response · Authors · 2024-11-26
> > >
> > > Thank you for your positive feedback! Your support means a lot to us.

---

### Official Review · Reviewer_W7XB · 2024-11-04

**Soundness:** 2
**Presentation:** 3
**Contribution:** 3
**Rating:** 6
**Confidence:** 3

**Summary:**

Merging multiple LLMs, particularly those with substantial parameter shifts from pre-training (PT), presents challenges for traditional merging methods. To address this issue, the paper introduces WIDEN (Weight Disentanglement), a novel approach for merging large language models (LLMs) that have undergone either fine-tuning (FT) or pre-training (PT). This method expands the applicability of model merging beyond conventional fine-tuned models.

**Strengths:**

1. The paper makes a valuable contribution by identifying a critical limitation in existing model merging methods: their ineffectiveness when applied to continually pre-trained (PT) models. This insight is essential, as it highlights a gap in current merging techniques, which are generally only effective for fine-tuned (FT) models with minimal parameter shifts.
2. The paper introduces WIDEN (Weight Disentanglement), an innovative method that automatically computes the importance of weights during the merging process. WIDEN disentangles each model’s weights into magnitude and direction components, and then adapts the merging decisions based on the divergence of these components from a shared backbone. This approach removes the need for manually assigning scaling factors and effectively addresses the challenges posed by the varied parameter changes in both fine-tuned (FT) and pre-trained (PT) models.
3. The experimental results demonstrate that WIDEN outperforms existing merging methods by effectively combining both instruction-following and multilingual capabilities. The paper also evaluates WIDEN in traditional FT-only merging scenarios, where it achieves competitive performance compared to established methods.

**Weaknesses:**

1. The paper assumes that continually pre-trained (PT) models inherently experience larger weight shifts than fine-tuned (FT) models, which serves as the justification for a new merging approach. However, this assumption may not hold universally, as the degree of weight change in PT models depends on factors such as the data domain and dataset size. This raises questions about the paper’s motivation and the general applicability of its problem formulation. A more thorough exploration or empirical verification of weight changes across PT and FT models would help substantiate this claim. The authors are expected to provide empirical evidence comparing the distribution of weight changes between PT and FT models across different domains, model sizes, and dataset sizes.
2. The proposed ranking mechanism in WIDEN calculates divergences in magnitude and direction separately for each weight relative to the backbone model. However, the reliability of comparing magnitudes across models with different directional vectors is questionable. When calculating magnitude differences, direction is not considered, meaning that the importance of weights in different models could be misinterpreted if their directions diverge. Similarly, comparing directional differences might be misleading if the corresponding magnitudes differ significantly between models. So, have the authors considered alternative approaches that jointly consider both magnitude and direction? Additionally, have the authors empirically analyzed how often misinterpretation occur in practice due to treating these components separately?
3. Although WIDEN is intended to be a general merging technique applicable to both FT and PT models, its performance in merging FT models is comparatively weak (as shown in Table 5). Given that the method is designed to be adaptable across model types, this underperformance raises concerns about its overall efficacy. Are there certain characteristics of FT models that WIDEN struggles with?
4. The experiments primarily focus on merging a specific PT model (Sailor) with a FT model, which limits the generalization ability of the results. Evaluating WIDEN on other PT models, particularly in diverse domains such as finance or healthcare, would provide stronger evidence of its effectiveness.

**Questions:**

1. Equation 1: The shape of mD is equal to that of W. This equation should be corrected.
2. Equation 7: Could you provide a reason for not averaging all the differences by multiplying by 1/N?
3. Have the authors considered an alternative approach that compares each weight matrix on a column-by-column basis between the tuned model and the original backbone? Specifically, this approach would involve calculating and ranking differences column by column, rather than disentangling weights into separate magnitude and direction components.
4. How to grid search the hyperparameters for baselines methods? What validation dataset is used?
5. The paper should provide a figure to visually illustrate the proposed method.

---

> ### Author Response · Authors · 2024-11-22
> **Response to Reviewer W7XB (Part 1/4)**
>
> Thanks for your constructive feedback. We have undertaken several revisions to address your comments. Firstly, we have investigated and discussed the parameter changed ranges of different LLMs across various domains, model sizes, and dataset sizes. Secondly, we have added merging experiments of Sailor and Qwen1.5-Chat at 1.8B and 4B model scales as well as pointed out the common issue of insufficient pre-training in most existing PT LLMs. Thirdly, we have elaborated on the importance of the weight disentanglement operation in WIDEN and verified its significance through an ablation study. Finally, we have provided a detailed explanation of the grid search process and included a framework figure of WIDEN for better comprehension. We hope these revisions have adequately addressed your concerns and we remain open to any additional questions or feedback.
>
> **W1: A more thorough exploration or empirical verification of weight changes across PT and FT models is desired. The authors are expected to provide empirical evidence comparing the distribution of weight changes between PT and FT models across different domains, model sizes, and dataset sizes.**
>
> We acknowledge the importance of thoroughly exploring weight changes across PT and FT models. In Table 10 of Appendix Section A.7 of the manuscript, we presented quantitative statistics on the parameter changed ranges for LLMs used in our merging experiments, specifically PT Sailor and FT Qwen1.5-Chat, with model sizes of 7B and 14B. The findings indicate that Sailor exhibits a significantly larger parameter changed range (about 0.008) compared to Qwen1.5-Chat (around 0.0004), which is approximately 1/20th of Sailor’s magnitude. This disparity is likely due to the difference in number of training tokens: Sailor undergoes 200B tokens during continued pre-training [1], whereas Qwen1.5-Chat was fine-tuned on significantly fewer tokens, with fine-tuning datasets for its successor (Qwen2-Instruct) reportedly consisting of around 0.5M samples [2].
>
> To further address your concern, we conduct additional investigations on several existing PT LLMs, including Sailor (1.8B and 4B), finance-chat, medicine-chat, law-chat (based on Llama-2-7b-chat) [3], BioMistral-7B (based on Mistral-7B-Instruct-v0.1) [4], and Saul-7B-Base (based on Mistral-7B-v0.1) [5]. These models are detailed in Appendix Sections A.6 (Table 9) and A.7 (Table 10). Our analysis reveals that most current PT LLMs (except for Sailor) are pre-trained on fewer than 30B tokens, resulting in relatively small parameter changed ranges. This makes them less suitable for our experimental setup, as substantial parameter changes among the models to be merged are desired. The above analysis has been included in Appendix Sections A.6 and A.7 in the revised manuscript.
>
> | **Models** | **Backbones** | **Domains** | **Training Tokens** |
> | --- | --- | --- | --- |
> | Sailor-1.8B | Qwen1.5-1.8B | Multilingual | 200B |
> | Sailor-4B | Qwen1.5-4B | Multilingual | 200B |
> | Sailor-7B | Qwen1.5-7B | Multilingual | 200B |
> | Sailor-14B | Qwen1.5-14B | Multilingual | 200B |
> | finance-chat | Llama-2-7b-chat | Finance Analysis | 1.2B |
> | medicine-chat | Llama-2-7b-chat | Medical Analysis | 5.4B |
> | law-chat | Llama-2-7b-chat | Law Assistance | 16.7B |
> | BioMistral-7B | Mistral-7B-Instruct-v0.1 | Medical Analysis | 3B |
> | Saul-7B-Base | Mistral-7B-v0.1 | Law Assistance | 30B |
>
> | **Models** | **0% (min)** | **10%** | **20%** | **30%** | **40%** | **50%** | **60%** | **70%** | **80%** | **90%** | **100% (max)** |
> | --- | --- | --- | --- | --- | --- | --- | --- | --- | --- | --- | --- |
> | Sailor-1.8B vs. Qwen1.5-1.8B | -6.25e-02 | -1.00e-02 | -0.51e-02 | -0.23e-02 | -0.06e-02 | 0.00 | 0.06e-02 | 0.23e-02 | 0.51e-02 | 1.00e-02 | 6.25e-02 |
> | Sailor-4B vs. Qwen1.5-4B | -0.63 | -0.96e-02 | -0.62e-02 | -0.38e-02 | -0.18e-02 | 0.00 | 0.18e-02 | 0.38e-02 | 0.62e-02 | 0.96e-02 | 0.63 |
> | Sailor-7B vs. Qwen1.5-7B | -0.27 | -0.57e-02 | -0.37e-02 | -0.23e-02 | -0.11e-02 | 0.00 | 0.11e-02 | 0.23e-02 | 0.37e-02 | 0.57e-02 | 0.25 |
> | Sailor-14B vs. Qwen1.5-14B | -0.36 | -0.78e-02 | -0.51e-02 | -0.31e-02 | -0.15e-02 | 0.00 | 0.15e-02 | 0.31e-02 | 0.51e-02 | 0.78e-02 | 0.42 |
> | finance-chat vs. Llama-2-7b-chat | -3.78e-02 | -3.66e-04 | -3.05e-05 | 0.00 | 0.00 | 0.00 | 0.00 | 0.00 | 3.05e-05 | 3.66e-04 | 5.07e-02 |
> | medicine-chat vs. Llama-2-7b-chat | -3.79e-02 | -0.03e-02 | 0.00 | 0.00 | 0.00 | 0.00 | 0.00 | 0.00 | 0.00 | 0.03e-02 | 5.03e-02 |
> | law-chat vs. Llama-2-7b-chat | -3.61e-02 | -0.03e-02 | 0.00 | 0.00 | 0.00 | 0.00 | 0.00 | 0.00 | 0.00 | 0.03e-02 | 4.77e-02 |
> | BioMistral-7B vs. Mistral-7B-Instruct-v0.1 | -6.25e-02 | -0.11e-02 | -0.07e-02 | -0.04e-02 | -0.02e-02 | 0.00 | 0.02e-02 | 0.04e-02 | 0.07e-02 | 0.11e-02 | 1.86e-02 |
> | Saul-7B-Base vs. Mistral-7B-v0.1 | -4.40e-03 | -1.22e-04 | -7.63e-05 | -4.58e-05 | -2.48e-05 | 0.00 | 2.48e-05 | 4.58e-05 | 7.63e-05 | 1.22e-04 | 4.15e-03 |

---

> > ### Author Response · Authors · 2024-11-22
> > **Response to Reviewer W7XB (Part 2/4)**
> >
> > [1] Dou L, Liu Q, Zeng G, et al. Sailor: Open Language Models for South-East Asia. 2024, arXiv.
> >
> > [2] Yang A, Yang B, Hui B, et al. Qwen2 technical report, 2024, arXiv.
> >
> > [3] Cheng D, Huang S, Wei F. Adapting large language models via reading comprehension. 2024, ICLR.
> >
> > [4] Labrak Y, Bazoge A, Morin E, et al. Biomistral: A collection of open-source pretrained large language models for medical domains. 2024, ACL Findings.
> >
> > [5] Colombo P, Pires T P, Boudiaf M, et al. Saullm-7b: A pioneering large language model for law. 2024, arXiv.
> >
> > **W2:  Have the authors considered to jointly consider both magnitude and direction or empirically analyze how often misinterpretation occur in practice due to treating these components separately?**
> >
> > The proposed WIDEN inherently considers both magnitude and direction. Specifically, Equations (2) through (6) compute the changes in the tuning model relative to the backbone model in terms of both magnitude and direction. Equation (7) then integrates these changes to determine the overall importance for model merging. Such a design ensures a holistic analysis of parameter changes and addresses the potential for misinterpretation when considering magnitude or direction separately. In case of “jointly consideration of both magnitude and direction” means comparing each weight matrix on a column-by-column basis, we provide an ablation study to validate the effectiveness of weight disentanglement in WIDEN. Results show that disentangling weight into magnitude and direction consistently performs better. Please see details on our response to **Q3**.
> >
> > **W3: Although WIDEN is intended to be a general merging technique applicable to both FT and PT models, its performance in merging FT models is comparatively weak. Are there certain characteristics of FT models that WIDEN struggles with?**
> >
> > We acknowledge that WIDEN performs competitively but less prominently than baselines when merging multiple FT models. This is because WIDEN excels at merging LLMs with obvious differences in parameter changed ranges by disentangling parameters into magnitudes and directions. In the case of FT models with minor and similar parameter changes (as shown in Table 10 in Appendix Section A.7), treating weights holistically or disentangling them leads to minimal disparity, which makes the disentanglement operation less pronounced. Kindly note that we report the performance of merging multiple FT LLMs to show that, in addition to the advantage of WIDEN in integrating PT and FT LLMs, WIDEN is also able to achieve competitive performance under a traditional experimental setup. We have incorporated the above analysis in Section 4.3 in the revised version.
> >
> > **W4: Evaluating WIDEN on other PT models, particularly in diverse domains such as finance or healthcare, would provide stronger evidence of its effectiveness.**
> >
> > Thank you for this insightful comment. Firstly, we have added results for Sailor and Qwen1.5-Chat with 1.8B and 4B parameters to validate the effectiveness of WIDEN across different model sizes. The results show that WIDEN performs well at both 1.8B and 4B sizes, effectively integrating the multilingual capabilities of Sailor while maintaining the general abilities of Qwen1.5-Chat. Moreover, the performance of WIDEN consistently improves with increasing model sizes on the Open LLM Leaderboard, indicating its potential scalability. These findings have been added to Section 4.2 and Appendix Section A.8 in the revised version.

---

> ### Author Response · Authors · 2024-11-22
> **Response to Reviewer W7XB (Part 3/4)**
>
> **Performance on the South-East Asian language benchmark:**
> | **Size**|**Models**|**Merging Methods**|**XQuAD (th)**|**TydiQA (id)**|**XQuAD (vi)**|**XCOPA (th)**|**XCOPA (id)**|**XCOPA (vi)**|**Belebele (th)**|**Belebele (id)**|**Belebele (vi)**|**M3Exam (jv)**|**Average**|**Average Rank**|
> | ---|---|---|---|---|---|---|---|---|---|---|---|---|---|---|
> | 1.8B|Qwen1.5|/|27.24/43.56|29.73/53.76|29.17/48.15|52.60|51.60|53.40|30.11|32.00|31.33|24.26|38.99|/|
> | 1.8B|Qwen1.5-Chat|/|18.10/31.43|24.42/49.10|24.64/43.13|53.00|53.20|54.40|29.89|32.00|34.00|26.15|36.42|/|
> | 1.8B|Sailor|/|32.72/48.66|40.88/65.37|34.22/53.35|53.80|64.20|63.20|34.22|34.89|35.33|28.30|45.32|/|
> | 1.8B|Qwen1.5-Chat & Sailor|Task Arithmetic|36.81/51.43|33.81/62.82|32.68/52.62|55.00|**65.40**|59.80|**34.33**|36.22|**36.11**|**28.30**|45.03|1.85|
> | 1.8B|Qwen1.5-Chat & Sailor|SLERP|28.37/44.64|21.77/53.76|29.26/51.39|54.40|54.40|57.40|32.22|34.33|35.44|27.22|40.35|4.15|
> | 1.8B|Qwen1.5-Chat & Sailor|Model Stock|28.63/44.35|30.97/56.50|31.65/51.14|52.80|51.60|54.80|30.89|33.00|31.44|23.99|40.14|4.85|
> | 1.8B|Qwen1.5-Chat & Sailor|Breadcrumbs|22.45/31.95|20.18/43.83|25.49/42.11|53.40|57.40|59.80|31.56|34.67|34.89|27.22|37.30|4.92|
> | 1.8B|Qwen1.5-Chat & Sailor|TIES-Merging|26.02/41.09|36.81/61.68|31.99/52.40|52.00|62.60|**60.40**|33.78|**36.89**|35.89|25.61|42.86|3.15|
> | 1.8B|Qwen1.5-Chat & Sailor|WIDEN|**38.21**/**53.50**|**43.36**/**68.55**|**37.55**/**56.05**|**55.20**|61.80|60.20|34.22|35.33|36.00|27.49|**46.73**|**1.62**|
> | 4B|Qwen1.5|/|34.03/53.40|48.32/72.68|43.71/63.86|53.40|55.00|57.80|32.78|36.22|35.22|24.26|46.98|/|
> | 4B|Qwen1.5-Chat|/|27.76/41.84|44.96/66.09|39.95/59.46|51.20|52.80|53.60|34.11|39.33|37.44|24.80|44.10|/|
> | 4B|Sailor|/|46.82/63.34|53.98/73.48|47.65/67.09|53.40|69.20|68.20|36.11|41.33|38.89|31.27|53.14|/|
> | 4B|Qwen1.5-Chat & Sailor|Task Arithmetic|**28.98**/**45.21**|16.28/28.27|19.76/36.27|53.80|60.40|58.40|34.11|39.11|36.89|23.99|37.04|2.85|
> | 4B|Qwen1.5-Chat & Sailor|SLERP|11.92/28.09|19.47/42.16|**31.74**/52.56|51.40|57.00|56.60|33.33|39.44|**38.22**|25.88|37.52|2.54|
> | 4B|Qwen1.5-Chat & Sailor|Model Stock|10.27/26.73|16.64/47.73|30.37/**52.69**|51.00|53.00|58.00|31.89|38.56|37.11|**27.22**|37.02|3.08|
> | 4B|Qwen1.5-Chat & Sailor|Breadcrumbs|0.70/1.80|5.49/9.14|1.54/1.67|48.80|56.20|55.80|28.33|29.11|30.56|24.80|22.61|4.92|
> | 4B|Qwen1.5-Chat & Sailor|TIES-Merging|0.00/0.50|0.18/2.86|0.43/1.13|52.00|53.00|52.80|26.44|29.56|29.11|24.53|20.96|5.46|
> | 4B|Qwen1.5-Chat & Sailor|WIDEN|25.67/45.08|**20.00**/**48.80**|25.49/42.17|**54.00**|**63.40**|**58.80**|**35.89**|**42.00**|33.22|24.53|**39.93**|**1.92**|
>
> **Performance on the Open LLM Leaderboard:**
> | **Size**|**Models**|**Merging Methods**|**ARC**|**HellaSwag**|**MMLU**|**TruthfulQA**|**Winogrande**|**GSM8K**|**Average**|**Average Rank**|
> | ---|---|---|---|---|---|---|---|---|---|---|
> | 1.8B|Qwen1.5|/|37.80|61.67|45.71|39.33|61.64|34.04|46.70|/|
> | 1.8B|Qwen1.5-Chat|/|39.68|60.36|44.53|40.57|59.83|31.39|46.06|/|
> | 1.8B|Sailor|/|32.59|57.48|29.60|37.77|59.98|2.65|36.68|/|
> | 1.8B|Qwen1.5-Chat & Sailor|Task Arithmetic|37.20|60.43|41.45|38.95|61.96|12.74|42.12|4.83|
> | 1.8B|Qwen1.5-Chat & Sailor|SLERP|**39.51**|61.17|43.96|**40.95**|60.85|25.40|45.31|2.17|
> | 1.8B|Qwen1.5-Chat & Sailor|Model Stock|37.97|**61.82**|**46.23**|39.84|61.96|**34.50**|**47.05**|**1.67**|
> | 1.8B|Qwen1.5-Chat & Sailor|Breadcrumbs|37.80|60.56|41.44|38.36|**62.04**|17.36|42.93|3.50|
> | 1.8B|Qwen1.5-Chat & Sailor|TIES-Merging|37.54|60.56|41.13|39.39|61.72|14.25|42.41|4.50|
> | 1.8B|Qwen1.5-Chat & Sailor|WIDEN|37.71|60.47|41.61|40.54|61.64|13.04|42.50|3.67|
> | 4B|Qwen1.5|/|48.04|71.43|55.01|47.22|68.43|52.31|57.07|/|
> | 4B|Qwen1.5-Chat|/|43.26|69.67|54.07|44.74|66.61|5.84|47.37|/|
> | 4B|Sailor|/|44.45|69.38|36.80|37.03|65.35|11.75|44.13|/|
> | 4B|Qwen1.5-Chat & Sailor|Task Arithmetic|46.50|64.01|38.25|43.73|65.19|8.49|44.36|4.00|
> | 4B|Qwen1.5-Chat & Sailor|SLERP|45.56|68.25|50.01|43.88|66.38|41.70|52.63|2.83|
> | 4B|Qwen1.5-Chat & Sailor|Model Stock|**47.01**|**69.31**|**55.41**|46.55|**67.32**|**47.08**|**55.45**|**1.33**|
> | 4B|Qwen1.5-Chat & Sailor|Breadcrumbs|39.16|43.15|43.84|48.55|61.80|0.00|39.42|4.33|
> | 4B|Qwen1.5-Chat & Sailor|TIES-Merging|35.15|41.04|30.15|**49.47**|59.19|0.00|35.83|5.00|
> | 4B|Qwen1.5-Chat & Sailor|WIDEN|45.90|66.05|48.66|43.34|66.69|13.95|47.43|3.33|
>
> Secondly, as noted in our response to **W1**, we observed that the pre-training process of most existing PT LLMs (such as finance-chat, medicine-chat, law-chat, BioMistral-7B, and Saul-7B-Base) is insufficient. As a result, their parameter changes are not significantly different from those of certain FT LLMs. This limitation makes it challenging to identify sufficiently pre-trained PT LLMs for further experiments during the rebuttal period. We hope the above explanations resolve your concern, and we plan to continue investigating these challenging scenarios in future work.

---

> > ### Author Response · Authors · 2024-11-22
> > **Response to Reviewer W7XB (Part 4/4)**
> >
> > **Q1: In Equation 1, the shape of $mD$ is equal to that of** $W$ **and should be corrected.**
> >
> > Thank you for highlighting this detail. In Equation (1), $m$ is calculated by performing a norm operation along the first dimension of size $d$, resulting in $m$ having shape $\mathbb{R}^{1 \times k}$. The shape of $D$ matches that of $W$, which is $\mathbb{R}^{d \times k}$. Therefore, the shape of $mD$ has the shape $\mathbb{R}^{d \times k}$. We have clarified the shape in Equation (1) in our revised version.
> >
> > **Q2: Could the authors provide a reason for not averaging all the differences by multiplying by $1/N$ in Equation 7?**
> >
> > We have already accounted for the importance of weights in different models through the normalization operation in Equation (4). Therefore, directly considering the normalized $\mathcal{M}$ and $\mathcal{D}$ in Equation (7) can sufficiently reflect the contributions of different models without the need to explicitly multiply by $1/N$. Compared to simply averaging all the differences by multiplying by $1/N$, our approach allows for a more fine-grained calculation of the importance of each weight across different models based on both magnitude and direction, providing greater flexibility. Additionally, as mentioned in Remark 2 of Section 3.3, multiplying the differences by $1/N$ (i.e., Average Merging) is actually a special case of WIDEN in extreme situations.
> >
> > **Q3: Have the authors considered an alternative approach that compares each weight matrix on a column-by-column basis between the tuned model and backbone? For example, calculating and ranking differences column by column, rather than disentangling weights into separate magnitude and direction components.**
> >
> > Yes, we have considered such an alternative approach. In Section 4.4 of our paper, we implemented a variant of WIDEN, referred to as **WIDEN w/o WD**, which calculates the discrepancy between the weights of the LLM and the corresponding backbone using cosine similarities, essentially the mentioned column-by-column approach. WIDEN w/o WD directly considers weights to compute weight importance instead of disentangling weights into magnitude and direction components. Results in Figure 2 show that on both 7B and 14B model sizes, WIDEN (in pink) consistently outperforms the non-disentangled WIDEN w/o WD (in green), validating the effectiveness of the weight disentanglement operation.
> >
> > **Q4: How to grid search the hyperparameters for baselines methods? What validation dataset is used?**
> >
> > For hyperparameter selection, we sample 10% of the data from each dataset in the benchmarks as the validation set for grid search. The settings that yield the best average performance on the validation set are selected for evaluation. This process is uniformly applied to all baseline methods as well as WIDEN to ensure a fair comparison. We acknowledge that a more rigorous approach might involve using entirely separate datasets for validation and testing. However, given the challenge of ensuring the validation set remains relevant to the diverse test set and justifying which datasets may have been encountered in the training data, we did not use additional external datasets for validation at this stage. We have added the above explanations in Appendix Section A.5 in the revised manuscript.
> >
> > **Q5: The paper should provide a figure to visually illustrate the proposed method.**
> >
> > Thanks for this helpful comment. We have included a framework diagram of the proposed WIDEN (Figure 4) in Appendix Section A.1. This visual aid is intended to enhance the understanding of the computational process of WIDEN.

---

> > > ### Comment · Reviewer_W7XB · 2024-11-25
> > > **Official Comment by Reviewer W7XB**
> > >
> > > Thank the authors for the detailed response. The responses have addressed most of my concerns. And I decide to raise my score.

---

> > > > ### Author Response · Authors · 2024-11-25
> > > >
> > > > Thank you very much for your positive feedback and support!

---

### Author Response · Authors · 2024-11-22
**General Response to All Reviewers**

We sincerely thank all the reviewers for their valuable feedback and insightful comments on our work. We are encouraged by the recognition that our research is **well-motivated** and **addresses the limitations of existing merging methods when applied to Pre-Trained (PT) Large Language Models (LLMs)** (Reviewers W7XB, KKeW, and 3uPt), the proposed WIDEN is **innovative** and **technically sound** (Reviewers W7XB and 3uPt), our **experimental results are effective, extensive, and thorough** (Reviewers W7XB, KKeW, and 3uPt), and the paper is **well-written and easy to follow** (Reviewer KKeW).

To the best of our efforts, we have diligently addressed each concern raised by the reviewers through the following enhancements:

- We explored parameter changed ranges across a variety of existing LLMs spanning different domains, architectures, and model sizes;
- We conducted further merging experiments on 1.8B and 4B PT Sailor and Fine-Tuned (FT) Qwen1.5-Chat;
- We discussed the characteristics of LLMs that are more suitable for WIDEN;
- We provided explanations for the varied performance gaps between WIDEN and baselines across different benchmarks;
- We attempted to merge three FT LLMs using WIDEN and compared its performance with baselines;
- We investigated the performance of WIDEN under different settings of hyperparameters $t$ and $s$;
- We elaborated on the importance of the weight disentanglement operation in WIDEN and provided details of the grid search process.

All these improvements are highlighted in blue in the revised manuscript. We think that all the comments are very constructive and believe that the revisions can enhance the quality of our work.

---

> ### Author Response · Authors · 2024-11-25
> **Gentle Reminder Regarding Rebuttal Response of Submission 1006**
>
> Dear Reviewers,
>
> Thank you for your time and valuable feedback on our submission. We have carefully addressed your comments and submitted our rebuttal response. We kindly request you to review our response at your earliest convenience and let us know if there are any further clarifications or questions.
>
> We appreciate your efforts in reviewing our work and look forward to hearing your thoughts.
>
> Best regards,
>
> Authors of submission 1006

---

### Meta-Review · Area_Chair_vaBV · 2024-12-21

**Metareview:**

This paper tackles the model merging problem between PT and FT LLMs. The approach focuses on merging homogeneous models that are of a significant parameter divergence. The basic idea is to separate the weight into the magnitude and the direction components to automatically determine the merging weights. The authors took the Sailor as the PT LLM and showcased positive results. However, this paper received some common and critical concerns, especially that the method is limited to significant parameter change scenarios and performs less effectively with FT model merging and other PT models beyond the Sailor series. After discussion, Reviewer W7XB improved the rating score from 5 to 6 but decreased the confidence score from 4 to 3. Other reviewers kept their scores unchanged. After reviewing all materials, the AC found that the critical concerns were not all addressed. Considering the overall rating, the final recommendation is reject.

**Additional Comments On Reviewer Discussion:**

This paper received three reviews. Some common concerns are about the moderated performance, missing evaluation on other PT models, missing analysis of merging FT models, and other concerns about the generalization and scalability.
After rebuttal, Reviewer W7XB thought the concerns were responded to and, therefore, improved the score from 5 to 6. However, the reviewer lowered the confidence score from 4 to 3.
Other reviewers kept their ratings unchanged, and reviewer 3uPt did not provide final comments.
After reading the paper and all the discussions, the AC found that the concerns about the model's generalization and scalability were not fully addressed. The method is limited to significant parameter change scenarios. It performs less effectively with FT model merging and other PT models beyond the Sailor series. Considering the overall rating and confidence scores, the final recommendation is reject.

---

### Decision · Program_Chairs · 2025-01-22

Reject